# Does Adversarial Transferability Indicate Knowledge Transferability?

## Abstract

Despite the immense success that deep neural networks (DNNs) have achieved, *adversarial examples*, which are perturbed inputs that aim to mislead DNNs to make mistakes, have recently led to great concerns. On the other hand, adversarial examples exhibit interesting phenomena, such as *adversarial transferability*. DNNs also exhibit knowledge transfer, which is critical to improving learning efficiency and learning in domains that lack high-quality training data. To uncover the fundamental connections between these phenomena, we investigate and give an affirmative answer to the question: *does adversarial transferability indicate knowledge transferability?* We theoretically analyze the relationship between adversarial transferability and knowledge transferability, and outline easily checkable sufficient conditions that identify when adversarial transferability indicates knowledge transferability. In particular, we show that composition with an affine function is sufficient to reduce the difference between the two models when they possess high adversarial transferability. Furthermore, we provide empirical evaluation for different transfer learning scenarios on diverse datasets, showing a strong positive correlation between the adversarial transferability and knowledge transferability, thus illustrating that our theoretical insights are predictive of practice.

## 1 Introduction

Knowledge transferability and adversarial transferability are two fundamental properties when a learned model transfers to other domains. Knowledge transferability, also known as learning transferability, has attracted extensive studies in machine learning. Long before it was formally defined, the computer vision community has exploited it to perform important visual manipulations (Johnson et al., 2016), such as style transfer and super-resolution, where pretrained VGG networks (Simonyan & Zisserman, 2014) are utilized to encode images into semantically meaningful features. After the release of ImageNet (Russakovsky et al., 2015), pretrained ImageNet models (e.g., on TensorFlow Hub or PyTorch-Hub) has quickly become the default option for the transfer source, because of its broad coverage of visual concepts and compatibility with various visual tasks (Huh et al., 2016). Adversarial transferability, on the other hand, is a phenomenon that adversarial examples can not only attack the model they are generated against, but also affect other models (Goodfellow et al., 2014; Papernot et al., 2016). Thus, adversarial transferability is extensively exploited to inspire black-box attacks (Ilyas et al., 2018; Liu et al., 2016). Many theoretical analyses have been conducted to establish sufficient conditions of adversarial transferability (Demontis et al., 2019; Ma et al., 2018).

Knowledge transferability and adversarial transferability both reveal some nature of machine learning models and the corresponding data distributions. Particularly, the relation between these two phenomena interests us the most. We begin by showing that adversarial transferability can indicate knowledge transferability. This tie can potentially provide a similarity measure between data distributions, an identifier of important features focused by a complex model, and an affinity map between complicated tasks. Thus, we believe our results have further implications in model interpretability and verification, fairness, robust and efficient transfer learning, and etc.

To the best of our knowledge, this is the first work studying the fundamental relationship between adversarial transferability and knowledge transferability both theoretically and empirically. Our main contributions are as follows.

- We formally define two quantities, $\tau_1$ and $\tau_2$, to *measure adversarial transferability from different aspects*, which enables in-depth understanding of adversarial transferability from a geometric point of view in the feature representation space.
- We derive an *upper bound for knowledge transferability with respect to adversarial transferability*. We rigorously depict their underlying relation and show that adversarial transferability can indicate knowledge transferability.
- We conduct thorough controlled experiments for diverse knowledge transfer scenarios (e.g. knowledge transfer among data distributions, attributes, and tasks) on benchmark datasets including STL-10, CIFAR-10, CelebA, Taskonomy-data, and four language datasets. Our empirical results show *strong positive correlation* between adversarial and knowledge transferability, which validates our theoretical prediction.

## 2 RELATED WORK

**Knowledge transferability** has been widely applied in scenarios where the available data for certain domain is limited, and has achieved great success (Van Opbroek et al., 2014; Wurm et al., 2019; Wang et al., 2017; Kim & Park, 2017; Maqueda et al., 2018; Devlin et al., 2018). Several studies have been conducted to understand the factors that affect knowledge transferability (Yosinski et al., 2014; Long et al., 2015b; Wang et al., 2019; Xu et al., 2019; Shinya et al., 2019). Empirical observations show that the correlation between learning tasks (Achille et al., 2019; Zamir et al., 2018), the similarity of model architectures, and data distribution are all correlated with different knowledge transfer effects.

**Adversarial Transferability** has been observed by several works (Papernot et al., 2016; Goodfellow et al., 2014; Joon Oh et al., 2017). Since the early work, a lot of studies have been conducted, aiming to further understand the phenomenon and design more transferable adversarial attacks. Regardless of the threat model, a lot of attack methods have been proposed to boost adversarial transferability (Zhou et al., 2018; Demontis et al., 2019; Dong et al., 2019; Xie et al., 2019). Naseer et al. (2019) propose to produce adversarial examples that transfer cross-domain via a generative adversarial network. In addition to the efficacy, efficiency (Ilyas et al., 2018) and practicality (Papernot et al., 2017) are also optimized. Beyond the above empirical studies, there is some work dedicated to analyzing this phenomenon, showing different conditions that may enhance adversarial transferability (Athalye et al., 2018; Tramèr et al., 2017; Ma et al., 2018; Demontis et al., 2019). Building upon these observations, it is clear that there exist certain connections between adversarial transferability and other knowledge transfer scenarios, and here we aim to provide the first theoretic justification to verify it and design systematic empirical studies to measure such correlation.

## 3 ADVERSARIAL TRANSFERABILITY VS. KNOWLEDGE TRANSFERABILITY

In this section, we establish connections between adversarial examples and knowledge transferability rigorously. We first formally state the problem studied in this section. Then, we move on to subsection 3.1 to introduce two metrics that encode information about adversarial attacks. Finally, we present our theoretical results about the relationship between adversarial and knowledge transferability in subsection 3.2.

**Notations.** We use blackboard bold to denote sets, e.g., $\mathbb{R}$. We use calligraphy to denote distributions, e.g., $\mathcal{D}$. The support of a distribution $\mathcal{D}$ is denoted as $\text{supp}(\mathcal{D})$. We use bold lower case letters to denote vectors, e.g., $\boldsymbol{x} \in \mathbb{R}^n$. We use bold uppercase letter to denote a matrix, e.g., $\boldsymbol{A}$. We use $\boldsymbol{A}^\dagger$ to denote the Moore–Penrose inverse of matrix $\boldsymbol{A}$. We use $\circ$ to denote the composition of functions, i.e., $g \circ f(\boldsymbol{x}) = g(f(\boldsymbol{x}))$. We use $\| \cdot \|_2$ to denote Euclidean norm induced by standard inner product $\langle \cdot, \cdot \rangle$. Given a function $f$, we use $f(\boldsymbol{x})$ to denote its evaluated value at $\boldsymbol{x}$, and we use $f$ to represent this function in function space. We use $\langle \cdot, \cdot \rangle_\mathcal{D}$ to denote inner product induced by distribution $\mathcal{D}$, i.e., $\langle f_1, f_2 \rangle_\mathcal{D} = \mathbb{E}_{\boldsymbol{x} \sim \mathcal{D}} \langle f_1(\boldsymbol{x}), f_2(\boldsymbol{x}) \rangle$. Accordingly, we use $\| \cdot \|_\mathcal{D}$ to denote a norm induced by inner product $\langle \cdot, \cdot \rangle_\mathcal{D}$, i.e., $\|f\|_\mathcal{D} = \sqrt{\langle f, f \rangle_\mathcal{D}}$. For a matrix function $F : \text{supp}(\mathcal{D}) \to \mathbb{R}^{d \times m}$, we define its $L^2(\mathcal{D})$-norm in accordance with matrix 2-norm as $\|F\|_{\mathcal{D},2} = \sqrt{\mathbb{E}_{\boldsymbol{x} \sim \mathcal{D}} \|F(\boldsymbol{x})\|_2^2}$. We define projection operator $\text{proj}(\cdot, r)$ to project a matrix to a hyperball of spectral norm radius $r$, i.e.,

$$\text{proj}(\boldsymbol{A}, r) = \begin{cases} \boldsymbol{A}, & \text{if} \quad \|\boldsymbol{A}\|_2 \leq r \\ r\boldsymbol{A}/\|\boldsymbol{A}\|_2 & \text{if} \quad \|\boldsymbol{A}\|_2 > r \end{cases}.$$

**Setting**. Assume we are given a target problem defined by data distribution $\boldsymbol{x} \sim \mathcal{D}$, where $\boldsymbol{x} \in \mathbb{R}^n$, and $y : \mathbb{R}^n \to \mathbb{R}^d$ represent the ground truth labeling function. As a first try, a reference model $f_T : \mathbb{R}^n \to \mathbb{R}^d$ trained on the target dataset is obtained through optimizing over a function class $f_T \in \mathbb{F}_T$. Now suppose we have a source model $f_S : \mathbb{R}^n \to \mathbb{R}^m$ pretrained on source data, and we are curious how would $f_S$ transfer to the target data $\mathcal{D}$?

**Knowledge transferability.** Given a trainable function $g : \mathbb{R}^m \to \mathbb{R}^d$, where $g \in \mathbb{G}$ is from a small function class for efficiency purpose, we care about whether $f_S$ can achieve low loss $\mathcal{L}(\cdot; y, \mathcal{D})$, e.g., mean squared error, after stacking with a trainable function $g$ comparing with $f_T$, i.e.,

$$\min_{g \in \mathbb{G}} \quad \mathcal{L}(g \circ f_S; y, \mathcal{D}) \quad \text{compare with} \quad \mathcal{L}(f_T; y, \mathcal{D}).$$

Clearly, the solution to this optimization problem depends on the choice of $\mathbb{G}$. Observing that in practice it is common to stack and fine-tune a linear layer given a pretrained feature extractor, we consider the class of affine functions. Formally, the problem that is studied in our theory is stated as follows.

**Problem 1.** *Given a reference model $f_T$ trained on target distribution $\mathcal{D}$, and a source model $f_S$ pre-trained on source data. Can we predict the best possible performance of the composite function $g \circ f_S$ on $\mathcal{D}$, where $g$ is from a bounded affine function class, given adversarial transferability between $f_S$ and $f_T$?*

### 3.1 Adversarial Transferability

We use the $\ell_2$-norm to characterize the effectiveness of an attack.

**Definition 1** (Virtual Adversarial Attack (Miyato et al., 2018)). *Given a model $f : \mathbb{R}^n \to \mathbb{R}^d$, the attack on point $\boldsymbol{x}$ within $\epsilon$-ball is defined as $\arg\max_{\|\boldsymbol{\delta}\| \leq \epsilon} \|f(\boldsymbol{x}) - f(\boldsymbol{x} + \boldsymbol{\delta})\|_2$. As this is intractable in practice, we consider the use of the tangent function to approximate the difference:*

$$\boldsymbol{\delta}_{f,\epsilon}(\boldsymbol{x}) \quad = \quad \arg\max_{\|\boldsymbol{\delta}\| \leq \epsilon} \|\nabla f(\boldsymbol{x})^\top \boldsymbol{\delta}\|_2,$$

*where $\nabla f(\boldsymbol{x}) \in \mathbb{R}^{n \times d}$ is the Jacobian matrix. The $\epsilon$ will be dropped in clear context or when it is irrelevant.*

To provide a quantitative view of adversarial transferability, we define two metrics $\tau_1$ and $\tau_2$. Both the metrics are in the range of $[0, 1]$, where higher values indicate more adversarial transferability.

**Definition 2** (Adversarial Transferability (Angle)). *Given two function $f_1, f_2$, we assume they have the same input dimension, and may have different output dimensions. The Adversarial Transferability (Angle) of $f_1$ and $f_2$ at point $\boldsymbol{x}$ is defined as the squared cosine value of the angle between the two attacks, i.e.,*

$$\tau_1(\boldsymbol{x}) = \frac{\langle \boldsymbol{\delta}_{f_1}(\boldsymbol{x}), \boldsymbol{\delta}_{f_2}(\boldsymbol{x}) \rangle^2}{\|\boldsymbol{\delta}_{f_1}(\boldsymbol{x})\|_2^2 \cdot \|\boldsymbol{\delta}_{f_2}(\boldsymbol{x})\|_2^2}.$$

*We denote its expected value as $\tau_1 = \mathbb{E}_{\boldsymbol{x} \sim \mathcal{D}}[\tau_1(\boldsymbol{x})]$.*

Intuitively, $\tau_1$ characterizes the similarity of the two attacks. The higher the cosine similarity, the better they can be attacked together. Noting that we are suggesting to use the square of their cosine values, which means that cosine value being either $1$ or $-1$ has the same indication of high knowledge transferability. This is because fine-tuning the last layer can rectify such difference by changing the sign of the last linear layer. However, it is not sufficient to fully characterize how good $f_S$ will perform only knowing the angle of two attack directions. For example, it is not difficult to construct two functions with highest $\tau_1 = 1$, but not transferable with affine functions. Moreover, it is also oberserved in our experiments that only $\tau_1$ is not sufficient.

Therefore, in addition to the information of attacks $\delta_f$ captured by $\tau_1$, we also need information about *deviation* of a function given attacks. We denote the deviation of a function $f$, given attack $\boldsymbol{\delta}(\boldsymbol{x})$, as $f(\boldsymbol{x} + \boldsymbol{\delta}(\boldsymbol{x})) - f(\boldsymbol{x})$, and we define its approximation as

$$\Delta_{f,\boldsymbol{\delta}}(\boldsymbol{x}) = \nabla f(\boldsymbol{x})^\top \boldsymbol{\delta}(\boldsymbol{x}). \tag{1}$$

Accordingly, we define another metric to answer the following question: applying $f_1$'s adversarial attacks on both the models, how much can the deviation of their function value be aligned by affine transformations?

**Definition 3** (Adversarial Transferability (Deviation)). *Given two functions $f_1$, $f_2$ with the same input dimensions and potentially different output dimensions, the Adversarial Transferability (Deviation) of adversarial attacks from $f_1$ to $f_2$ given data distribution $\mathcal{D}$ is defined as*

$$\tau_2^{f_1 \to f_2} = \frac{\langle 2\Delta_{f_2, \boldsymbol{\delta}_{f_1}} - \boldsymbol{A}\Delta_{f_1, \boldsymbol{\delta}_{f_1}}, \boldsymbol{A}\Delta_{f_1, \boldsymbol{\delta}_{f_1}} \rangle_{\mathcal{D}}}{\|\Delta_{f_2, \boldsymbol{\delta}_{f_1}}\|_{\mathcal{D}}^2},$$

*where $\boldsymbol{A}$ is a constant matrix defined as*

$$\boldsymbol{A} = proj(\mathbb{E}_{\boldsymbol{x} \sim \mathcal{D}}[\Delta_{f_2, \boldsymbol{\delta}_{f_1}}(\boldsymbol{x})\Delta_{f_1, \boldsymbol{\delta}_{f_1}}(\boldsymbol{x})^\top] \left(\mathbb{E}_{\boldsymbol{x} \sim \mathcal{D}}[\Delta_{f_1, \boldsymbol{\delta}_{f_1}}(\boldsymbol{x})\Delta_{f_1, \boldsymbol{\delta}_{f_1}}(\boldsymbol{x})^\top]\right)^\dagger, \frac{\|\Delta_{f_2, \boldsymbol{\delta}_{f_1}}\|_{\mathcal{D}}}{\|\Delta_{f_1, \boldsymbol{\delta}_{f_1}}\|_{\mathcal{D}}}).$$

We note that $\boldsymbol{A}$ is the best linear map trying to align the two deviations ($\Delta_{f_2, \boldsymbol{\delta}_{f_1}}$ and $\Delta_{f_1, \boldsymbol{\delta}_{f_1}}$) in the function space. It serves as a guess on the best linear map to align $f_1$ and $f_2$, using only the information from adversarial attacks. To have better sense of $\tau_2$ and the relationships with other quantities, we present an example for visual illustration in Figure 1. Note that high $\tau_2$ does not necessarily require $\Delta_{f_1, \boldsymbol{\delta}_{f_1}}$ and $\Delta_{f_2, \boldsymbol{\delta}_{f_1}}$ to be similar, but they can be well aligned by the constant linear transformation $\boldsymbol{A}$. We refer to the proof of Proposition 1 at section B in appendix for detailed explanation of $\tau_2$.

**Proposition 1.** *Both $\tau_1$ and $\tau_2$ are in $[0, 1]$.*

## 3.2 ADVERSARIAL TRANSFERABILITY INDICATES KNOWLEDGE TRANSFERABILITY

In this subsection, we will provide our theoretical results. First, to have a better intuition, we will show a special case where the theorems are simplified, i.e., where $f_S$ and $f_T$ are both $\mathbb{R}^n \to \mathbb{R}$. Then, we present the general case where $f_S$ and $f_T$ are multi-dimensional. Note that their output dimensions are not necessarily the same.

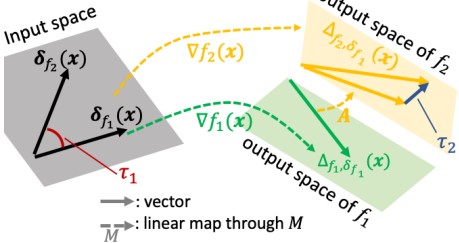

Figure 1: Illustration of the key variables.

When $f_S$ and $f_T$ are both $\mathbb{R}^n \to \mathbb{R}$, the $\tau_1$ and $\tau_2$ come out in a surprisingly elegant form. Let us show what the two metrics are to have further intuition on what $\tau_1$ and $\tau_2$ characterize.

First, let us see what the attack is in this case. As function $f$ has one-dimensional output, its gradient is a vector $\nabla f \in \mathbb{R}^n$. Thus,

$$\boldsymbol{\delta}_{f, \epsilon}(x) = \arg\max_{\|\boldsymbol{\delta}\| \le \epsilon} \|\nabla f(\boldsymbol{x})^\top \boldsymbol{\delta}\|_2 = \frac{\epsilon \nabla f(\boldsymbol{x})}{\|\nabla f(\boldsymbol{x})\|_2}$$

is simply the gradient with its scale normalized. Then, the $\tau_1$ becomes

$$\tau_1(\boldsymbol{x}) = \frac{\langle \nabla f_S(\boldsymbol{x}), \nabla f_T(\boldsymbol{x}) \rangle^2}{\|\nabla f_S(\boldsymbol{x})\|_2^2 \cdot \|\nabla f_T(\boldsymbol{x})\|_2^2},$$

which is the squared cosine (angle) between two gradients. For $\tau_2$, the matrix $\boldsymbol{A}$ degenerates to a scalar constant, which makes $\tau_2$ simpler as well, i.e.,

$$A = \frac{\langle \Delta_{f_T, \boldsymbol{\delta}_{f_S}}, \Delta_{f_S, \boldsymbol{\delta}_{f_S}} \rangle_{\mathcal{D}}}{\|\Delta_{f_S, \boldsymbol{\delta}_{f_S}}\|_{\mathcal{D}}^2}, \quad \text{and} \quad \tau_2^{f_S \to f_T} = \frac{\langle \Delta_{f_S, \boldsymbol{\delta}_{f_S}}, \Delta_{f_T, \boldsymbol{\delta}_{f_S}} \rangle_{\mathcal{D}}^2}{\|\Delta_{f_S, \boldsymbol{\delta}_{f_S}}\|_{\mathcal{D}}^2 \cdot \|\Delta_{f_T, \boldsymbol{\delta}_{f_S}}\|_{\mathcal{D}}^2}.$$

We can see, in this case $\tau_2$ is interestingly in the same form of the first metric $\tau_1$. We will simply use $\tau_2$ to denote $\tau_2^{f_S \to f_T}$ afterwards.

Accordingly, when $f_S$ and $f_T$ are both $\mathbb{R}^n \to \mathbb{R}$, the result also comes out in an elegant form. In this case, adversarial attacks reflect all the information of the gradients of the two models, enabling $\tau_1$ and $\tau_2$ to encode all the information we need to prove the following theorem.

**Theorem 1.** *For two functions $f_S$ and $f_T$ that both are $\mathbb{R}^n \to \mathbb{R}$, there is an affine function $g : \mathbb{R} \to \mathbb{R}$, such that*

$$\|\nabla f_T - \nabla(g \circ f_S)\|_{\mathcal{D}}^2 = \mathbb{E}_{\boldsymbol{x} \sim \mathcal{D}}\left[(1 - \tau_1(\boldsymbol{x})\tau_2)\|\nabla f_T(\boldsymbol{x})\|_2^2\right],$$

where $g(x) = Ax + Const$. Moreover, though not necessarily, if assuming that $f_T$ is $L$-Lipschitz continuous, i.e., $\|\nabla f_T(\boldsymbol{x})\|_2 \leq L$ for $\forall \boldsymbol{x} \in supp(\mathcal{D})$, we have a more elegant statement:

$$\|\nabla f_T - \nabla(g \circ f_S)\|_{\mathcal{D}}^2 \leq (1 - \tau_1 \tau_2)L^2.$$

The theorem suggests that, if adversarial transferability is high, there exists an affine transformation with bounded norm, such that $g \circ f_S$ is close to $f_T$. As an intuition of the proof, the difference between two gradients can be represented by the angle between them, which can be characterized by $\tau_1$; and the norm difference between them, which can be characterized by $\tau_2$.

As for the general case, we consider when the output dimensions of both functions are multi-dimensional and not necessarily the same. In this scenario, adversarial attacks correspond to the largest singular value of the Jacobian matrix. Therefore, we need to introduce the following definition to capture other information that is not revealed by adversarial attacks.

**Definition 4** (Singular Value Ratio). *For any function $f$, the Singular Value Ratio for the function gradient at $\boldsymbol{x}$ is defined as $\lambda_f(\boldsymbol{x}) = \frac{\sigma_2(\boldsymbol{x})}{\sigma_1(\boldsymbol{x})}$, where $\sigma_1(\boldsymbol{x}), \sigma_2(\boldsymbol{x})$ are the largest and the second largest singular value in absolute value of $\nabla f(\boldsymbol{x})$, respectively. In addition, we define the worst-case singular value ratio as $\lambda_f = \max_{\boldsymbol{x} \in supp(\mathcal{D})} \lambda_f(\boldsymbol{x})$.*

**Theorem 2.** *For two functions $f_S : \mathbb{R}^n \to \mathbb{R}^m$, and $f_T : \mathbb{R}^n \to \mathbb{R}^d$, assuming that $f_T$ is $L$-Lipschitz continuous, i.e., $\|\nabla f_T(\boldsymbol{x})\|_2 \leq L$ for $\forall \boldsymbol{x} \in supp(\mathcal{D})$, there is an affine function $g : \mathbb{R}^m \to \mathbb{R}^d$, such that*

$$\|\nabla f_T - \nabla(g \circ f_S)\|_{\mathcal{D}}^2 \leq \left( (1 - \tau_1 \tau_2) + (1 - \tau_1)(1 - \tau_2)\lambda_{f_T}^2 + (\lambda_{f_T} + \lambda_{f_S})^2 \right) 5L^2,$$

*where $g$ is defined as $g(\boldsymbol{z}) = \boldsymbol{A}\boldsymbol{z} + \boldsymbol{Const}$.*

We note that this theorem also has a statement offering tighter bound where we do not assume Lipschitz continuous. The full version of this theorem is provided in appendix. Theorem 2 suggests that big $\tau_1$ and $\tau_2$ indicate potentially small differences of gradients between the target model and the transferred model. Based on this, intuitively, given the right constant value shift, minimal difference in gradients implies minimal difference in function value, which should result in bounded loss. Indeed, we prove in Theorem 3 that the squared loss of the transferred model $g \circ f_S$ is bounded by the loss of $f_T$ and their gradient difference, by assuming the $\beta$-smoothness of both the functions.

**Definition 5** ($\beta$-smoothness). *A function $f$ is $\beta$-smooth if for all $\boldsymbol{x}, \boldsymbol{y}$,*

$$\|\nabla f(\boldsymbol{x}) - \nabla f(\boldsymbol{y})\|_2 \leq \beta\|\boldsymbol{x} - \boldsymbol{y}\|_2.$$

For the target data distribution $\mathcal{D}$, and its ground truth labeling function $y$, the mean squared loss of the transferred model is $\mathbb{E}_{\boldsymbol{x} \sim \mathcal{D}}\|g \circ f_S(\boldsymbol{x}) - y(\boldsymbol{x})\|_2^2 = \|g \circ f_S - y\|_{\mathcal{D}}^2$. Therefore, the following theorem presents upper bound on the mean squared loss of the transferred model.

**Theorem 3.** *Without loss of generality we assume $\|\boldsymbol{x}\|_2 \leq 1$ for $\forall \boldsymbol{x} \in supp(\mathcal{D})$. Consider functions $f_S : \mathbb{R}^n \to \mathbb{R}^m$, $f_T : \mathbb{R}^n \to \mathbb{R}^d$, and an affine function $g : \mathbb{R}^m \to \mathbb{R}^d$, suggested by Theorem 1 or Theorem 2, with the constant set to let $g(f_S(\boldsymbol{0})) = f_T(\boldsymbol{0})$. If both $f_T, f_S$ are $\beta$-smooth, then*

$$\|g \circ f_S - y\|_{\mathcal{D}}^2 \leq \left( \|f_T - y\|_{\mathcal{D}} + \|\nabla f_T - \nabla g \circ f_S\|_{\mathcal{D}} + \left(1 + \frac{\|\nabla f_T\|_{\mathcal{D},2}}{\|\nabla f_S\|_{\mathcal{D},2}}\right)\beta \right)^2.$$

### 3.3 PRACTICAL MEASUREMENT OF ADVERSARIAL TRANSFERABILITY

Existing studies have shown that similar models share high adversarial transferability (Liu et al., 2016; Papernot et al., 2016; Tramèr et al., 2017). In previous work, it is common to use cross adversarial loss as an indication of adversarial transferability, e.g., the loss of $f_T$ with attacks generated on $f_S$. It is intuitive to consider that the higher cross adversarial loss, the higher adversarial transferability. However, it may have a drawback comparing to the $\tau_1, \tau_2$ defined in this work.

**Definition 6** (Cross Adversarial Loss). *Given a loss function $\ell_T(\cdot, y)$ on the target domain, where $y$ is ground truth, the adversarial loss of $f_T$ with attack $\boldsymbol{\delta}_{f_S}$ generated against source model $f_S$ is*

$$\mathcal{L}_{adv}(f_T, \boldsymbol{\delta}_{f_S}; y, \mathcal{D}) = \mathbb{E}_{\boldsymbol{x} \sim \mathcal{D}} \quad \ell_T(f_T(\boldsymbol{x} + \boldsymbol{\delta}_{f_S}(\boldsymbol{x})), y(\boldsymbol{x})).$$

The cross adversarial loss depends on the choice of loss function, the output dimension, etc. Thus, it can be incomparable when we want to test adversarial transferability among different $f_T$, unlike that $\tau_1, \tau_2$ are always between $[0, 1]$. To investigate the relationship between the adversarial loss and the adversarial transferability we defined, we show in the following proposition that the cross adversarial loss is similar to $\tau_1$. In the next section, we verify the theoretical predictions through thorough experiments.

**Proposition 2.** *If $\ell_T$ is mean squared loss and $f_T$ achieves zero loss on $\mathcal{D}$, then the adversarial loss defined in Definition 6 is approximately upper and lower bounded by*

$$\mathcal{L}_{adv}(f_T, \boldsymbol{\delta}_{f_S}; y, \mathcal{D}) \geq \epsilon^2 \mathbb{E}_{\boldsymbol{x} \sim \mathcal{D}} \left[ \tau_1(\boldsymbol{x}) \left\| \nabla f_T(\boldsymbol{x}) \right\|_2^2 \right] + O(\epsilon^3),$$

$$\mathcal{L}_{adv}(f_T, \boldsymbol{\delta}_{f_S}; y, \mathcal{D}) \leq \epsilon^2 \mathbb{E}_{\boldsymbol{x} \sim \mathcal{D}} \left[ \left( \lambda_{f_T}^2 + (1 - \lambda_{f_T}^2)\tau_1(\boldsymbol{x}) \right) \left\| \nabla f_T(\boldsymbol{x}) \right\|_2^2 \right] + O(\epsilon^3),$$

*where $O(\epsilon^3)$ denotes a cubic error term.*

## 4 EXPERIMENTAL EVALUATION

The empirical evaluation of the relationship between adversarial transferability and knowledge transferability is done by four different sets of experiment. First we present a set of synthetic experiment that verifies our theoretical study, and then we present our empirical study on real-world datasets with models widely used in practice, described in three knowledge transfer scenarios: knowledge transfer on data distributions, attributes, and tasks. Details regarding the three scenarios are elaborated below, and all training details are deferred to the Appendix.

**Knowledge-transfer among data distributions** is the most common setting of transfer learning. It transfers the knowledge of a model trained/gained from one data domain to the other data domains. For instance, Shie et al. (2015) manage to use pre-trained ImageNet representations to achieve state-of-the-art accuracy for medical data analysis. The relation between adversarial and knowledge transferability can not only determine the best pretrained models to use, but also detect distribution shifts, which is crucial in learning agents deployed in continual setting (Diethe et al., 2019).
**Knowledge-transfer among attributes** is a popular method to handle zero-shot and few-shot learning (Jayaraman & Grauman, 2014; Romera-Paredes & Torr, 2015). It transfers the knowledge learned from the attributes of the source problem to a new target problem Russakovsky & Fei-Fei (2010). The relation between adversarial and knowledge transferability can be used as a probe to deployed classification models to verify attributes that their decisions are based on. This will have profound implications on fairness and interpretability.
**Knowledge-transfer among tasks** is widely applied across various vision tasks, such as super resolution (Johnson et al., 2016), style transfer (Gatys et al., 2016), semantic and instance segmentation (Girshick, 2015; He et al., 2017; Long et al., 2015a). It involves transferring the knowledge the model gains by learning to do one task to another novel task. The relation between adversarial and knowledge transferability, as many recent works (Achille et al., 2019; Standley et al., 2019; Zamir et al., 2018), can be used to charting the affinity map between tasks, aiming to guide potential transfer.

### 4.1 SYNTHETIC EXPERIMENT ON RADIAL BASIS FUNCTIONS REGRESSION

In the synthetic experiment, we compute quantities that are otherwise inefficient to compute to verify our theoretical results. We also try different settings to see how other factors affect the results. Details follow.

**Models.** Both the source model $f_S$ and the target model $f_T$ are one-hidden-layer neural networks with sigmoid activation.

**Overall Steps.** First, sample $D = \{(\boldsymbol{x}_i, \boldsymbol{y}_i)\}_{i=1}^N$ from a distribution (details later), where $\boldsymbol{x}$ is $n$-dimensional, $\boldsymbol{y}$ is $d$-dimensional, and there are $N$ samples. Then we train a target model $f_T$ on $D$. Denoting the weights of $f_T$ as $\boldsymbol{W}$, we randomly sample a direction $\boldsymbol{V}$ where each entry of $\boldsymbol{V}$ is sampled from $U(-0.5, 0.5)$, and choose a scale $t \in [0, 1]$. To derive the source model, we perturb the target model as $\boldsymbol{W}' := \boldsymbol{W} + t\boldsymbol{V}$. Define the source model $f_S$ to be a one-hidden-layer neural network with weights $\boldsymbol{W}'$. Then, we compute each of the quantities we care about, including $\tau_1, \tau_2$, cross adversarial loss (Definition 6), the upper bound in theorem 2 on the difference of gradients, etc.

Noting that we reported the cross adversarial loss normalized by its own adversarial loss, defined as $\alpha = \|\Delta_{f_T, \delta_{f_S}}\|_{\mathcal{D}}^2 / \|\Delta_{f_T, \delta_{f_T}}\|_{\mathcal{D}}^2 \approx \mathcal{L}_{adv}(f_T, \delta_{f_S}; y, \mathcal{D}) / \mathcal{L}_{adv}(f_T, \delta_{f_T}; y, \mathcal{D})$ when $f_T$ achieves low error. Note that $\alpha \in [0, 1]$. Finally, we fine-tune the last layer of $f_S$, and get the true transferred loss.

**Dataset.** Denote a radial basis function as $\phi_i(\boldsymbol{x}) = e^{-\|\boldsymbol{x} - \boldsymbol{\mu}_i\|_2^2 / (2\sigma_i)^2}$, and we set the target ground truth function to be the sum of $M = 100$ basis functions as $f = \sum_{i=1}^M \phi_i$, where each entry of the parameters are sampled once from $U(-0.5, 0.5)$. We set the dimension of $\boldsymbol{x}$ to be 30, and the dimension of $\boldsymbol{y}$ to be 10. We generate $N = 1200$ samples of $\boldsymbol{x}$ from a Gaussian mixture formed by three Gaussian with different centers but the same covariance matrix $\boldsymbol{\Sigma} = \boldsymbol{I}$. The centers are sampled randomly from $U(-0.5, 0.5)^n$. We use the ground truth regressor $f$ to derive the corresponding $\boldsymbol{y}$ for each $\boldsymbol{x}$. That is, we want our neural networks to approximate $f$ on the Gaussian mixture.

**Results.** We present two sets of experiment in Figure 2. The correlations between adversarial transferabilities ($\tau_1, \tau_2, \alpha$) and the knowledge transferability (transferred loss) are observed. The upper bound for the difference of gradients (Theorem 2) basically tracks its true value. Although the absolute value of the upper bound on the transferred loss (Theorem 3) can be big compared to the true transferred loss, their trends are similar. We note the big difference in absolute value is due to the use of $\beta$-smoothness, which considers the worst case scenario. It is also observed that $\tau_1$ tracks the normalized adversarial cross loss $\alpha$, as Proposition 2 suggests.

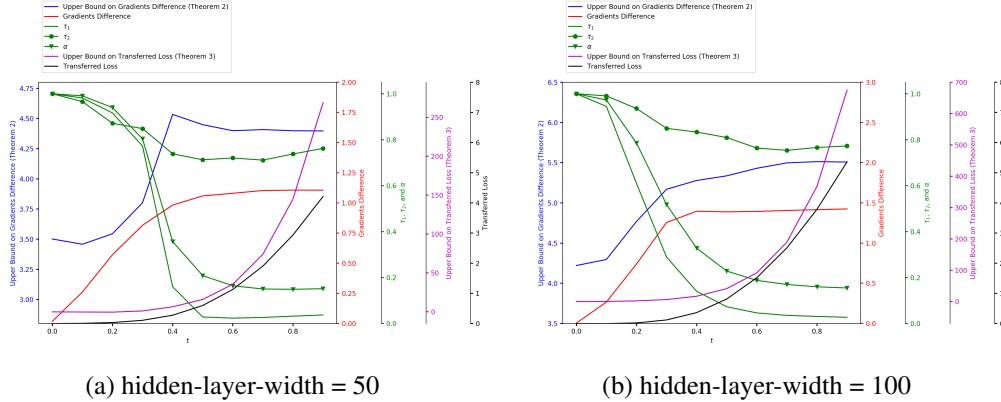

(a) hidden-layer-width = 50         (b) hidden-layer-width = 100

Figure 2: The x-axis is the $t \in [0, 1]$ that controls how much the source model deviates from the target model. There are in total 7 quantities reported, placed under 4 y-axes. Specifically, $\tau_1, \tau_2$, and the normalized cross adversarial loss $\alpha$ are plotted as green curves with green y-axis; the upper bound in theorem 2 on the transferred gradients difference is shown as blue curves with blue y-axis; the true transferred gradients difference is shown as red curves with red y-axis; the upper bound in theorem 3 on the transferred loss is shown as magenta curves with magenta y-axis; the true transferred loss is shown as **black** curves with **black** y-axis.

## 4.2 ADVERSARIAL TRANSFERABILITY INDICATES KNOWLEDGE-TRANSFER AMONG DATA DISTRIBUTIONS

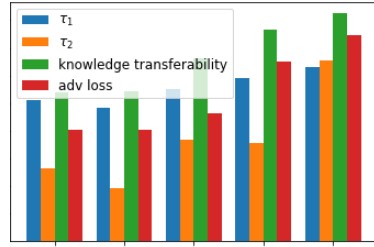

| Data | Yelp | AG | Fake |
|---|---|---|---|
| $\tau_1$ | 0.49 | 0.47 | 0.50 |
| $\tau_2$ | 1.34e-3 | 8e-4 | 3e-6 |
| adv loss | 0.16 | 0.13 | 0.12 |
| knowledge-trans | 0.89 | 0.66 | 0.52 |

Figure 3: (left) correlation between the adversarial transferability and knowledge trransferability in image domain (All values normalized to [0,1]). (right) adversarial transferability and knowledge transferability in NLP domain.

In this experiment, we show that the closer the source data distribution is to the target data distribution, the more adversarially transferable the source model to the reference model, thus we observe that the source model is more knowledge transferable to the target dataset. We demonstrate this on both image and natural language domains.

**Dataset.** Image: 5 source datasets (5 source models) are constructed based on CIFAR-10 (Hinton et al., 2012) and a single target dataset (1 reference model) based on STL-10 (Coates et al., 2011). Each of the source datasets consists of 4 classes from CIFAR10 and the target dataset also consists of 4 classes from STL10. Natural Language: We select 4 diverse natural language datasets: AG's News (AG), Fake News Detection (Fake), IMDB, Yelp Polarity (Yelp). Then we pick IMDB as the target and the rest as sources.

**Adversarial Transferability.** Image: We take 1000 images (STL-10) from the target dataset and generate 1000 adversarial examples on each of the five source models. We run 10 step PGD $L_\infty$ attack with $\epsilon = 0.1$. Then we measure the effectiveness of the adversarial examples by the cross-entropy loss on the reference model. Natural Language: We take 100 sample sentences from target dataset(IMDB) and generate adversarial sentences on each of the source models(AG, Fake, Yelp) with TextFooler(Jin et al., 2019). The ratio of changed words is constrained to less or equal to 0.1. Then, we measure their adversarial transferability against the reference model(IMDB).

**Knowledge Transferability.** To measure the knowledge transferability, we fine-tune a new linear layer on the target dataset to replace the last layer of each source model to generate the corresponding *transferred models*. Then we measure the performance of the transferred models on the target dataset based on the standard accuracy and cross-entropy loss.

**Results** From Figure 4.2, it's clear that if the source models that has highest adversarial transferability, its corresponding transferred model achieves the highest transferred accuracy. This phenomenon is prominent in both image and natural language domains. The results in Figure 4.2 (b) could verify the implication by our theory that only $\tau_1$ is not sufficient for indicating knowledge transferability.

| Data | Young | Male | Attractive | Eyebrows | Lipstick |
|:---:|:---:|:---:|:---:|:---:|:---:|
| $\tau_1$ | 0.0707 | 0.0679 | 0.0612 | 0.0609 | 0.0678 |
| $\tau_2$ | 0.0759 | 0.0521 | 0.0418 | 0.0529 | 0.0388 |
| adv loss | 17.83 | 16.21 | 14.13 | 13.22 | 12.54 |
| knowledge-trans | 0.593 | 0.589 | 0.562 | 0.551 | 0.554 |

Table 1: Top 5 Attributes with the highest adversarial transferability and their corresponding average accuracy on the validation benchmarks.

## 4.3 ADVERSARIAL TRANSFERABILITY INDICATING KNOWLEDGE-TRANSFER AMONG ATTRIBUTES

In addition to the data distributions, we validate our theory on another dimension, attributes. This experiment suggests that the more adversarially transferable the source model of certain attributes is to the reference model, the better the model performs on the target task aiming to learn target attributes.

**Dataset** CelebA (Liu et al., 2018) consists of 202,599 face images from 10,177 identities. A reference facial recognition model is trained on this identities. Each image also comes with 40 binary attributes, on which we train 40 source models. Our goal is to test whether source models of *source attributes*, can transfer to perform facial recognition.

**Adversarial Transferability** We sample 1000 images from CelebA and perform a virtual adversarial attack as described in section 3 on each of the 40 attribute classifiers. Then we measure the adversarial transfer effectiveness of these adversarial examples on the reference facial recognition model.

**Knowledge Transferability** To fairly assess the knowledge transferability, we test the 40 *transferred models* on 7 well-known facial recognition benchmarks, LFW (Huang et al., 2007), CFP-FF, CFP-FP (S. Sengupta, 2016), AgeDB (Moschoglou et al., 2017), CALFW, CPLFW (Zheng et al., 2017) and VGG2-FP (Cao et al., 2018). We report the average classification accuracy target datasets.

**Result** In Table 1, we list the top-5 attribute source models that share the highest adversarial transferability and the performance of their transferred models on the 7 target facial recognition benchmarks. We observe that the attribute "Young" has the highest adversarial transferability; as a result, it also achieves highest classification average performance across the 7 benchmarks.

## 4.4 Adversarial Transferability Indicating Knowledge-transfer among Tasks

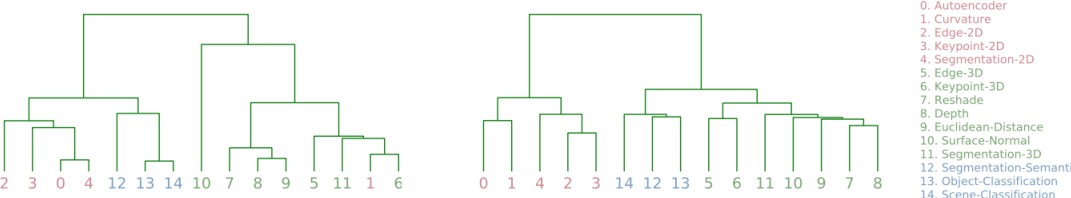

Figure 4: Left: Emprically confirmed taskonomy prediction of task categories (Zamir et al., 2018). Right: Task category prediction based on adversarial transferability. Different colors represent different task categories including 2D, 3D, Semantic. It is obvious that the adversarial transferability is able to predict similar task categories aligned with the pure knowledge-transfer empirical observation.

In this experiment, we show that adversarial transferability can also indicate the knolwdge transferability among different machine learning tasks. Zamir et al. (2018) shows that models trained on different tasks can transfer to other tasks well, especially when the tasks belong to the same "category". Here we leverage the same dataset, and pick 15 single image tasks from the task pool, including Autoencoding, 2D Segmentation, 3D Keypoint and etc. Intuitively, these tasks can be categorized into 3 categories, semantic task, 2D tasks as well as 3D tasks. Leveraging the tasks within the same category, which would hypothetically have higher adversarial transferability, we evaluate the corresponding knowledge transferability.

**Dataset** The Taskonomy-data consists of 4 million images of indoor scenes from about 600 indoor images, every one of which has annotations for every task listed in the pool. We use a public subset of these images to validate our theory.

**Adversarial Transferability** *Adversarial Transferability Matrix (ATM)* is used here to measure the adversarial transferability between multiple tasks, modified from the *Affinity Matrix* (Zamir et al., 2018). To generate the corresponding "task categories" for comparison, we sample 1000 images from the public dataset and perform a virtual adversarial attack on each of the 15 source models. Adversarial perturbation with $\epsilon$ ($L_\infty$ norm) as 0.03,0.06 are used and we run 10 steps PGD-based attack for efficiency. Detailed settings about adversarial transferability are deferred to the Appendix.

**Knowledge Transferability** We use the affinity scores provided as a $15 \times 15$ affinity matrix to compute the categories of tasks. Then we take columns of this matrix as features for each task and perform agglomerative clustering to obtain the Task Similarity Tree.

**Results** Figure 4 compares the predictions of task categories generated based on adversarial transferability and knowledge transferability in Taskonomy. It is easy to see three intuitive categories are formed, i.e, 2D, 3D, and Semantic tasks for both adversarial and knowledge transferability. To provide a quantitative measurement of the similarity, we also compute the average inner category entropy based on adversarial transferability with the categories in Taskonomy as the ground truth (the lower entropy indicates higher correlation between adversarial and knowledge transferability). In figure 5 (Appendix), the adversarial transferability based category prediction shows low entropy when the number of categories is greater or equal to 3, which indicates that the adversarial tranferability is faithful with the category prediction in Taskonomy. This result shows strong positive correlation between the adversarial transferability and knowledge transferability among learning tasks in terms of predicting the similar task categories.

## 5 Conclusion

We theoretically analyze the relationship between adversarial transferability and knowledge transferability, along with thorough experimental justifications in diverse scenarios. Both our theoretical and empirical results show that adversarial transferability can indicate knowledge transferability, which reveal important properties of machine learning models. We hope our discovery can inspire and facilitate further investigations, including model interpretability, fairness, robust and efficient transfer learning, and etc.

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

## A  DISCUSSION ABOUT VALIDNESS OF THE NOTATIONS

Before starting proving our theory, it is necessary to show that our mathematical tools are indeed valid. It is easy to verify that $\langle \cdot, \cdot \rangle_{\mathcal{D}}$ is a valid inner product inherited form standard Euclidean inner product. Therefore, the norm $\| \cdot \|_{\mathcal{D}}$, induced by the inner product, is also a valid norm.

What does not come directly is the validness of the norm $\| \cdot \|_{\mathcal{D},2}$. Particularly, whether it satisfies the triangle inequality. Recall that, for a function of matrix output $F : \text{supp}(\mathcal{D}) \to \mathbb{R}^{d \times m}$, its $L^2(\mathcal{D})$-norm in accordance with matrix 2-norm is defined as

$$\|F\|_{\mathcal{D},2} = \sqrt{\mathbb{E}_{\boldsymbol{x} \sim \mathcal{D}} \|F(\boldsymbol{x})\|_2^2}.$$

For two functions $F, G$, both are $\text{supp}(\mathcal{D}) \to \mathbb{R}^{d \times m}$, we can verify the norm $\| \cdot \|_{\mathcal{D},2}$ satisfies triangle inequality as shown in the following. Applying the triangle inequality of the spectral norm, and with some algebra manipulation, it holds that

$$
\begin{aligned}
\|F + G\|_{\mathcal{D},2} &= \sqrt{\mathbb{E}_{\boldsymbol{x} \sim \mathcal{D}} \|F(\boldsymbol{x}) + G(\boldsymbol{x})\|_2^2} \\
&\leq \sqrt{\mathbb{E}_{\boldsymbol{x} \sim \mathcal{D}} \left( \|F(\boldsymbol{x})\|_2 + \|G(\boldsymbol{x})\|_2 \right)^2} \\
&= \sqrt{\mathbb{E}_{\boldsymbol{x} \sim \mathcal{D}} \|F(\boldsymbol{x})\|_2^2 + \mathbb{E}_{\boldsymbol{x} \sim \mathcal{D}} \|G(\boldsymbol{x})\|_2^2 + 2\mathbb{E}_{\boldsymbol{x} \sim \mathcal{D}} \|F(\boldsymbol{x})\|_2 \|G(\boldsymbol{x})\|_2} \\
&= \sqrt{\|F\|_{\mathcal{D},2}^2 + \|G\|_{\mathcal{D},2}^2 + 2\mathbb{E}_{\boldsymbol{x} \sim \mathcal{D}} \|F(\boldsymbol{x})\|_2 \|G(\boldsymbol{x})\|_2}.
\end{aligned}
\tag{2}
$$

Applying the Cauchy-Schwarz inequality, we can see that

$$
\begin{aligned}
\mathbb{E}_{\boldsymbol{x} \sim \mathcal{D}} \|F(\boldsymbol{x})\|_2 \|G(\boldsymbol{x})\|_2 &\leq \sqrt{\mathbb{E}_{\boldsymbol{x} \sim \mathcal{D}} \|F(\boldsymbol{x})\|_2^2 \cdot \mathbb{E}_{\boldsymbol{x} \sim \mathcal{D}} \|G(\boldsymbol{x})\|_2^2} \\
&= \|F\|_{\mathcal{D},2} \cdot \|G\|_{\mathcal{D},2}.
\end{aligned}
$$

Plugging this into (2) would complete the proof, i.e.,

$$
\begin{aligned}
(2) &\leq \sqrt{\|F\|_{\mathcal{D},2}^2 + \|G\|_{\mathcal{D},2}^2 + 2\|F\|_{\mathcal{D},2} \cdot \|G\|_{\mathcal{D},2}} \\
&= \sqrt{(\|F\|_{\mathcal{D},2} + \|G\|_{\mathcal{D},2})^2} \\
&= \|F\|_{\mathcal{D},2} + \|G\|_{\mathcal{D},2}.
\end{aligned}
$$

## B  PROOF OF PROPOSITION 1

**Proposition 1** (Restated). *Both $\tau_1$ and $\tau_2$ are in $[0, 1]$.*

*Proof.* We are to prove that $\tau_1$ and $\tau_2$ are both in the range of $[0, 1]$. As $\tau_1$ is squared cosine, it is trivial that $\tau_1 \in [0, 1]$. Therefore, we will focus on $\tau_2$ in the following.

Recall that the $\tau_2$ from $f_1$ to $f_2$ is defined as

$$\tau_2^{f_1 \to f_2} = \frac{\langle 2\Delta_{f_2, \boldsymbol{\delta}_{f_1}} - \boldsymbol{A}\Delta_{f_1, \boldsymbol{\delta}_{f_1}}, \boldsymbol{A}\Delta_{f_1, \boldsymbol{\delta}_{f_1}} \rangle_{\mathcal{D}}}{\|\Delta_{f_2, \boldsymbol{\delta}_{f_1}}\|_{\mathcal{D}}^2},$$

where $\boldsymbol{A}$ is a constant matrix defined as

$$\boldsymbol{A} = \text{proj}(\mathbb{E}_{\boldsymbol{x} \sim \mathcal{D}}[\Delta_{f_2, \boldsymbol{\delta}_{f_1}}(\boldsymbol{x}) \Delta_{f_1, \boldsymbol{\delta}_{f_1}}(\boldsymbol{x})^\top] \left(\mathbb{E}_{\boldsymbol{x} \sim \mathcal{D}}[\Delta_{f_1, \boldsymbol{\delta}_{f_1}}(\boldsymbol{x}) \Delta_{f_1, \boldsymbol{\delta}_{f_1}}(\boldsymbol{x})^\top]\right)^\dagger, \frac{\|\Delta_{f_2, \boldsymbol{\delta}_{f_1}}\|_{\mathcal{D}}}{\|\Delta_{f_1, \boldsymbol{\delta}_{f_1}}\|_{\mathcal{D}}}).$$

For notation convenience, we will simply use $\tau_2$ to denote $\tau_2^{f_1 \to f_2}$ in this proof.

$\tau_2$ characterizes how similar are the changes in both the function values of $f_1 : \mathbb{R}^n \to \mathbb{R}^m$ and $f_2 : \mathbb{R}^n \to \mathbb{R}^d$ in the sense of linear transformable, given attack generated on $f_1$. That is being said, it is associated to the function below, i.e,

$$h(\boldsymbol{B}) = \left\|\Delta_{f_2, \boldsymbol{\delta}_{f_1}} - \boldsymbol{B}\Delta_{f_1, \boldsymbol{\delta}_{f_1}}\right\|_{\mathcal{D}}^2 = \mathbb{E}_{\boldsymbol{x} \sim \mathcal{D}} \left\|\Delta_{f_2, \boldsymbol{\delta}_{f_1}}(\boldsymbol{x}) - \boldsymbol{B}\Delta_{f_1, \boldsymbol{\delta}_{f_1}}(\boldsymbol{x})\right\|_2^2,$$

where $\Delta_{f_1,\delta_{f_1}} \in \mathbb{R}^m$, $\Delta_{f_2,\delta_{f_1}} \in \mathbb{R}^d$, and $\boldsymbol{B} \in \mathbb{R}^{d \times m}$.

As $\left\|\Delta_{f_2,\delta_{f_1}}(\boldsymbol{x}) - \boldsymbol{B}\Delta_{f_1,\delta_{f_1}}(\boldsymbol{x})\right\|_2^2$ is convex with respect to $\boldsymbol{B}$, its expectation, i.e. $h(\boldsymbol{B})$, is also convex.

Therefore, $h(\boldsymbol{B})$ it achieves global minima when $\frac{\partial h}{\partial \boldsymbol{B}} = 0$.

$$
\begin{aligned}
\frac{\partial h}{\partial \boldsymbol{B}} &= \mathbb{E}_{\boldsymbol{x} \sim \mathcal{D}} \frac{\partial}{\partial \boldsymbol{B}} \left( \left\|\Delta_{f_2,\delta_{f_1}}(\boldsymbol{x}) - \boldsymbol{B}\Delta_{f_1,\delta_{f_1}}(\boldsymbol{x})\right\|_2^2 \right) \\
&= 2\mathbb{E}_{\boldsymbol{x} \sim \mathcal{D}} \left[ \left(\boldsymbol{B}\Delta_{f_1,\delta_{f_1}}(\boldsymbol{x}) - \Delta_{f_2,\delta_{f_1}}(\boldsymbol{x})\right) \Delta_{f_1,\delta_{f_1}}(\boldsymbol{x})^\top \right] \\
&= 2\mathbb{E}_{\boldsymbol{x} \sim \mathcal{D}} \left[ \boldsymbol{B}\Delta_{f_1,\delta_{f_1}}(\boldsymbol{x})\Delta_{f_1,\delta_{f_1}}(\boldsymbol{x})^\top - \Delta_{f_2,\delta_{f_1}}(\boldsymbol{x})\Delta_{f_1,\delta_{f_1}}(\boldsymbol{x})^\top \right] \\
&= 2\boldsymbol{B}\mathbb{E}_{\boldsymbol{x} \sim \mathcal{D}} \left[ \Delta_{f_1,\delta_{f_1}}(\boldsymbol{x})\Delta_{f_1,\delta_{f_1}}(\boldsymbol{x})^\top \right] - 2\mathbb{E}_{\boldsymbol{x} \sim \mathcal{D}} \left[ \Delta_{f_2,\delta_{f_1}}(\boldsymbol{x})\Delta_{f_1,\delta_{f_1}}(\boldsymbol{x})^\top \right].
\end{aligned}
$$

Letting $\frac{\partial h}{\partial \boldsymbol{B}} = 0$, and denoting the solution as $\boldsymbol{B}^*$, we have

$$
\boldsymbol{B}^* = \mathbb{E}_{\boldsymbol{x} \sim \mathcal{D}} \left[ \Delta_{f_2,\delta_{f_1}}(\boldsymbol{x})\Delta_{f_1,\delta_{f_1}}(\boldsymbol{x})^\top \right] \left( \mathbb{E}_{\boldsymbol{x} \sim \mathcal{D}} \left[ \Delta_{f_1,\delta_{f_1}}(\boldsymbol{x})\Delta_{f_1,\delta_{f_1}}(\boldsymbol{x})^\top \right] \right)^\dagger .
$$

Noting that $\boldsymbol{A} = \text{proj}(\boldsymbol{B}^*, \frac{\|\Delta_{f_2,\delta_{f_1}}\|_\mathcal{D}}{\|\Delta_{f_1,\delta_{f_1}}\|_\mathcal{D}})$ is scaled $\boldsymbol{B}^*$, we denote $\boldsymbol{A} = \psi\boldsymbol{B}^*$, where $\psi$ a scaling factor depending on $\boldsymbol{B}^*$ and $\frac{\|\Delta_{f_2,\delta_{f_1}}\|_\mathcal{D}}{\|\Delta_{f_1,\delta_{f_1}}\|_\mathcal{D}}$. According to the definition of the projection operator, we can see that $0 < \psi \leq 1$.

Replacing $\boldsymbol{B}$ by $\boldsymbol{A}$ we have,

$$
\begin{aligned}
h(\boldsymbol{A}) &= \left\|\Delta_{f_2,\delta_{f_1}} - \boldsymbol{A}\Delta_{f_1,\delta_{f_1}}\right\|_\mathcal{D}^2 = \langle \Delta_{f_2,\delta_{f_1}} - \boldsymbol{A}\Delta_{f_1,\delta_{f_1}}, \Delta_{f_2,\delta_{f_1}} - \boldsymbol{A}\Delta_{f_1,\delta_{f_1}} \rangle_\mathcal{D} \\
&= \left\|\Delta_{f_2,\delta_{f_1}}\right\|_\mathcal{D}^2 - \langle 2\Delta_{f_2,\delta_{f_1}} - \boldsymbol{A}\Delta_{f_1,\delta_{f_1}}, \boldsymbol{A}\Delta_{f_1,\delta_{f_1}} \rangle_\mathcal{D} \\
&= (1 - \tau_2) \left\|\Delta_{f_2,\delta_{f_1}}\right\|_\mathcal{D}^2 .
\end{aligned}
$$

It is obvious that $h(\boldsymbol{A}) = \left\|\Delta_{f_2,\delta_{f_1}} - \boldsymbol{A}\Delta_{f_1,\delta_{f_1}}\right\|_\mathcal{D}^2 \geq 0$, thus we have $\tau_2 \leq 1$.

As for the lower bound for $\tau_2$, we will need to use properties of $\boldsymbol{B}$. Denoting $\boldsymbol{O}$ as an all-zero matrix, it holds that

$$
h(\boldsymbol{B}^*) = \min_{\boldsymbol{B}}\{h(\boldsymbol{B})\} \leq h(\boldsymbol{O}). \tag{3}
$$

For $\boldsymbol{A} = \psi\boldsymbol{B}^*$, according to the convexity of $h(\cdot)$ and the fact that $\psi \in [0, 1]$, we can see the following, i.e.,

$$
h(\boldsymbol{A}) = h(\psi\boldsymbol{B}^*) = h(\psi\boldsymbol{B}^* + (1 - \psi)\boldsymbol{O}) \leq \psi h(\boldsymbol{B}^*) + (1 - \psi)h(\boldsymbol{O}).
$$

Applying (3) to the above, we can see that

$$
h(\boldsymbol{A}) \leq h(\boldsymbol{O}).
$$

Noting that $h(\boldsymbol{A}) = (1 - \tau_2) \left\|\Delta_{f_2,\delta_{f_1}}\right\|_\mathcal{D}^2$ and $h(\boldsymbol{O}) = \left\|\Delta_{f_2,\delta_{f_1}}\right\|_\mathcal{D}^2$, the above inequality suggests that

$$
(1 - \tau_2) \left\|\Delta_{f_2,\delta_{f_1}}\right\|_\mathcal{D}^2 \leq \left\|\Delta_{f_2,\delta_{f_1}}\right\|_\mathcal{D}^2,
$$
$$
0 \leq \tau_2.
$$

Therefore, $\tau_2$ is upper bounded by 1 and lower bounded by 0. □

## C  PROOF OF THEOREM 1

Before actually proving the theorem, let us have a look at what $\tau_1$ and $\tau_2$ are in the case where $f_S$ and $f_T$ are both $\mathbb{R}^n \to \mathbb{R}$. In this case, both $\tau_1$ and $\tau_2$ come out in an elegant form. Let us show what the two metrics are to have further intuition on what $\tau_1$ and $\tau_2$ characterize.

First, let us see what the attack is in this case. As function $f$ has one-dimensional output, its gradient is a vector $\nabla f \in \mathbb{R}^n$. Thus,

$$\boldsymbol{\delta}_{f,\epsilon}(x) \quad = \quad \underset{\|\boldsymbol{\delta}\| \leq \epsilon}{\arg\max} \|\nabla f(\boldsymbol{x})^\top \boldsymbol{\delta}\|_2 = \frac{\epsilon \nabla f(\boldsymbol{x})}{\|\nabla f(\boldsymbol{x})\|_2}$$

Then, the $\tau_1$ becomes

$$\tau_1(\boldsymbol{x}) = \frac{\langle \nabla f_S(\boldsymbol{x}), \nabla f_T(\boldsymbol{x})\rangle^2}{\|\nabla f_S(\boldsymbol{x})\|_2^2 \cdot \|\nabla f_T(\boldsymbol{x})\|_2^2}$$

which is the squared cosine (angle) between two gradients.

For $\tau_2$, the matrix $\boldsymbol{A}$ degenerates to a scalar constant

$$A = \frac{\langle \Delta_{f_T, \boldsymbol{\delta}_{f_S}}, \Delta_{f_S, \boldsymbol{\delta}_{f_S}}\rangle_{\mathcal{D}}}{\|\Delta_{f_S, \boldsymbol{\delta}_{f_S}}\|_{\mathcal{D}}^2},$$

and the second metric becomes

$$\tau_2^{f_S \to f_T} = \frac{\langle \Delta_{f_S, \boldsymbol{\delta}_{f_S}}, \Delta_{f_T, \boldsymbol{\delta}_{f_S}}\rangle_{\mathcal{D}}^2}{\|\Delta_{f_S, \boldsymbol{\delta}_{f_S}}\|_{\mathcal{D}}^2 \cdot \|\Delta_{f_T, \boldsymbol{\delta}_{f_S}}\|_{\mathcal{D}}^2}$$

We can see, it is interestingly in the same form of the first metric $\tau_1$. We will simply use $\tau_2$ to denote $\tau_2^{f_S \to f_T}$ afterwards.

**Theorem 1** (Restated). *For two functions $f_S$ and $f_T$ that both are $\mathbb{R}^n \to \mathbb{R}$, there is an affine function $g : \mathbb{R} \to \mathbb{R}$, so that*

$$\|\nabla f_T - \nabla(g \circ f_S)\|_{\mathcal{D}}^2 = \mathbb{E}_{\boldsymbol{x} \sim \mathcal{D}} \left[ (1 - \tau_1(\boldsymbol{x})\tau_2)\|\nabla f_T(\boldsymbol{x})\|_2^2 \right],$$

*where $g$ is defined as $g(x) = Ax + Const$.*

*Moreover, if assuming that $f_T$ is $L$-Lipschitz continuous, i.e., $\|\nabla f_T(\boldsymbol{x})\|_2 \leq L$ for $\forall \boldsymbol{x} \in supp(\mathcal{D})$, we can have a more elegant statement:*

$$\|\nabla f_T - \nabla(g \circ f_S)\|_{\mathcal{D}}^2 \leq (1 - \tau_1\tau_2)L^2.$$

*Proof.* In the case where $g$ is a one-dimensional affine function, we write is as $g(z) = Az + b$, where $A$ is defined in the definition of $\tau_2$ (Definition 3). In this case, it enjoys a simple form of

$$A = \frac{\langle \Delta_{f_T, \boldsymbol{\delta}_{f_S}}, \Delta_{f_S, \boldsymbol{\delta}_{f_S}}\rangle_{\mathcal{D}}}{\|\Delta_{f_S, \boldsymbol{\delta}_{f_S}}\|_{\mathcal{D}}^2}.$$

Then, we can see that

$$\begin{aligned}
\|\nabla f_T - \nabla(g \circ f_S)\|_{\mathcal{D}}^2 &= \|\nabla f_T - A\nabla f_S\|_{\mathcal{D}}^2 \\
&= \mathbb{E}_{\boldsymbol{x} \sim \mathcal{D}} \left[ \|\nabla f_T(\boldsymbol{x}) - A\nabla f_S(\boldsymbol{x})\|_2^2 \right].
\end{aligned} \tag{4}$$

To continue, we split $\nabla f_T$ as two terms, i.e., one on the direction on $\nabla f_S$ and one orthogonal to $\nabla f_S$.

Denoting $\phi(\boldsymbol{x})$ as the angle between $\nabla f_T(\boldsymbol{x})$ and $\nabla f_S(\boldsymbol{x})$ in Euclidean space, we have

$$\begin{aligned}
\nabla f_T(\boldsymbol{x}) &= \cos(\phi(\boldsymbol{x}))\frac{\|\nabla f_T(\boldsymbol{x})\|_2}{\|\nabla f_S(\boldsymbol{x})\|_2}\nabla f_S(\boldsymbol{x}) + \nabla f_T(\boldsymbol{x}) - \cos(\phi(\boldsymbol{x}))\frac{\|\nabla f_T(\boldsymbol{x})\|_2}{\|\nabla f_S(\boldsymbol{x})\|_2}\nabla f_S(\boldsymbol{x}) \\
&= \cos(\phi(\boldsymbol{x}))\frac{\|\nabla f_T(\boldsymbol{x})\|_2}{\|\nabla f_S(\boldsymbol{x})\|_2}\nabla f_S(\boldsymbol{x}) + \boldsymbol{v}(\boldsymbol{x}),
\end{aligned} \tag{5}$$

where we denote $\boldsymbol{v}(\boldsymbol{x}) = \nabla f_T(\boldsymbol{x}) - \cos(\phi(\boldsymbol{x}))\frac{\|\nabla f_T(\boldsymbol{x})\|_2}{\|\nabla f_S(\boldsymbol{x})\|_2}\nabla f_S(\boldsymbol{x})$ for notation convenience.

We can see that $\boldsymbol{v}(\boldsymbol{x})$ is orthogonal to $\nabla f_S(\boldsymbol{x})$, thus $\|\boldsymbol{v}(\boldsymbol{x})\|_2 = \sqrt{1 - \cos^2(\phi(\boldsymbol{x}))}\|\nabla f_T(\boldsymbol{x})\|_2$. Recall that actually $\tau_1(\boldsymbol{x}) = \cos^2(\phi(\boldsymbol{x}))$, it can be written as $\|\boldsymbol{v}(\boldsymbol{x})\|_2 = \sqrt{1 - \tau_1(\boldsymbol{x})}\|\nabla f_T(\boldsymbol{x})\|_2$.

Then, plugging (5) into (4) we have

$$
\begin{aligned}
(4) &= \mathbb{E}_{\boldsymbol{x}\sim\mathcal{D}}\left[\|\cos(\phi(\boldsymbol{x}))\frac{\|\nabla f_T(\boldsymbol{x})\|_2}{\|\nabla f_S(\boldsymbol{x})\|_2}\nabla f_S(\boldsymbol{x}) + \boldsymbol{v}(\boldsymbol{x}) - A\nabla f_S(\boldsymbol{x})\|_2^2\right] \\
&= \mathbb{E}_{\boldsymbol{x}\sim\mathcal{D}}\left[\left\|\left(\cos(\phi(\boldsymbol{x}))\frac{\|\nabla f_T(\boldsymbol{x})\|_2}{\|\nabla f_S(\boldsymbol{x})\|_2} - A\right)\nabla f_S(\boldsymbol{x}) + \boldsymbol{v}(\boldsymbol{x})\right\|_2^2\right] \\
&= \mathbb{E}_{\boldsymbol{x}\sim\mathcal{D}}\left[\left\|\left(\cos(\phi(\boldsymbol{x}))\frac{\|\nabla f_T(\boldsymbol{x})\|_2}{\|\nabla f_S(\boldsymbol{x})\|_2} - A\right)\nabla f_S(\boldsymbol{x})\right\|_2^2 + \|\boldsymbol{v}(\boldsymbol{x})\|_2^2\right] \\
&= \mathbb{E}_{\boldsymbol{x}\sim\mathcal{D}}\left[\left\|\left(\cos(\phi(\boldsymbol{x}))\frac{\|\nabla f_T(\boldsymbol{x})\|_2}{\|\nabla f_S(\boldsymbol{x})\|_2} - A\right)\nabla f_S(\boldsymbol{x})\right\|_2^2 + (1 - \tau_1(\boldsymbol{x}))\|\nabla f_T(\boldsymbol{x})\|_2^2\right] \\
&= \mathbb{E}_{\boldsymbol{x}\sim\mathcal{D}}\left[\left\|\left(\cos(\phi(\boldsymbol{x}))\frac{\|\nabla f_T(\boldsymbol{x})\|_2}{\|\nabla f_S(\boldsymbol{x})\|_2} - A\right)\nabla f_S(\boldsymbol{x})\right\|_2^2\right] + \mathbb{E}_{\boldsymbol{x}\sim\mathcal{D}}(1 - \tau_1(\boldsymbol{x}))\|\nabla f_T(\boldsymbol{x})\|_2^2 \\
&= \mathbb{E}_{\boldsymbol{x}\sim\mathcal{D}}\left[\left(\cos(\phi(\boldsymbol{x}))\frac{\|\nabla f_T(\boldsymbol{x})\|_2}{\|\nabla f_S(\boldsymbol{x})\|_2} - A\right)^2\|\nabla f_S(\boldsymbol{x})\|_2^2\right] + \mathbb{E}_{\boldsymbol{x}\sim\mathcal{D}}(1 - \tau_1(\boldsymbol{x}))\|\nabla f_T(\boldsymbol{x})\|_2^2.
\end{aligned}
\tag{6}
$$

Now let us deal with the first term by plugging in

$$
A = \frac{\langle\Delta_{f_T,\boldsymbol{\delta}_{f_S}}, \Delta_{f_S,\boldsymbol{\delta}_{f_S}}\rangle_\mathcal{D}}{\|\Delta_{f_S,\boldsymbol{\delta}_{f_S}}\|_\mathcal{D}^2},
$$

where $\Delta_{f_T,\boldsymbol{\delta}_{f_S}}(\boldsymbol{x}) = \epsilon\cos(\phi(\boldsymbol{x}))\|\nabla f_T(\boldsymbol{x})\|_2$ and $\Delta_{f_S,\boldsymbol{\delta}_{f_S}}(\boldsymbol{x}) = \epsilon\|\nabla f_S(\boldsymbol{x})\|_2$, and we have

$$
\begin{aligned}
\mathbb{E}_{\boldsymbol{x}\sim\mathcal{D}}&\left(\cos(\phi(\boldsymbol{x}))\frac{\|\nabla f_T(\boldsymbol{x})\|_2}{\|\nabla f_S(\boldsymbol{x})\|_2} - A\right)^2\|\nabla f_S(\boldsymbol{x})\|_2^2 \\
&= \mathbb{E}_{\boldsymbol{x}\sim\mathcal{D}}\left(\cos(\phi(\boldsymbol{x}))\|\nabla f_T(\boldsymbol{x})\|_2 - A\|\nabla f_S(\boldsymbol{x})\|_2\right)^2 \\
&= \frac{1}{\epsilon^2}\mathbb{E}_{\boldsymbol{x}\sim\mathcal{D}}\left(\Delta_{f_T,\boldsymbol{\delta}_{f_S}}(\boldsymbol{x}) - A\Delta_{f_S,\boldsymbol{\delta}_{f_S}}(\boldsymbol{x})\right)^2 \\
&= \frac{1}{\epsilon^2}\mathbb{E}_{\boldsymbol{x}\sim\mathcal{D}}\left(\Delta_{f_T,\boldsymbol{\delta}_{f_S}}(\boldsymbol{x})^2 + A^2\Delta_{f_S,\boldsymbol{\delta}_{f_S}}(\boldsymbol{x})^2 - 2A\Delta_{f_T,\boldsymbol{\delta}_{f_S}}(\boldsymbol{x})\Delta_{f_S,\boldsymbol{\delta}_{f_S}}(\boldsymbol{x})\right) \\
&= \frac{1}{\epsilon^2}\left(\left\|\Delta_{f_T,\boldsymbol{\delta}_{f_S}}\right\|_\mathcal{D}^2 + A^2\left\|\Delta_{f_S,\boldsymbol{\delta}_{f_S}}\right\|_\mathcal{D}^2 - 2A\langle\Delta_{f_T,\boldsymbol{\delta}_{f_S}}, \Delta_{f_S,\boldsymbol{\delta}_{f_S}}\rangle_\mathcal{D}\right) \\
&= \frac{1}{\epsilon^2}\left(\left\|\Delta_{f_T,\boldsymbol{\delta}_{f_S}}\right\|_\mathcal{D}^2 + \frac{\langle\Delta_{f_T,\boldsymbol{\delta}_{f_S}}, \Delta_{f_S,\boldsymbol{\delta}_{f_S}}\rangle_\mathcal{D}^2}{\|\Delta_{f_S,\boldsymbol{\delta}_{f_S}}\|_\mathcal{D}^2} - 2\frac{\langle\Delta_{f_T,\boldsymbol{\delta}_{f_S}}, \Delta_{f_S,\boldsymbol{\delta}_{f_S}}\rangle_\mathcal{D}^2}{\|\Delta_{f_S,\boldsymbol{\delta}_{f_S}}\|_\mathcal{D}^2}\right) \\
&= \frac{\left\|\Delta_{f_T,\boldsymbol{\delta}_{f_S}}\right\|_\mathcal{D}^2}{\epsilon^2}\left(1 - \frac{\langle\Delta_{f_T,\boldsymbol{\delta}_{f_S}}, \Delta_{f_S,\boldsymbol{\delta}_{f_S}}\rangle_\mathcal{D}^2}{\|\Delta_{f_S,\boldsymbol{\delta}_{f_S}}\|_\mathcal{D}^2 \cdot \|\Delta_{f_T,\boldsymbol{\delta}_{f_S}}\|_\mathcal{D}^2}\right) \\
&= (1 - \tau_2)\mathbb{E}_{\boldsymbol{x}\sim\mathcal{D}}\left[\cos^2(\boldsymbol{x})\|\nabla f_T(\boldsymbol{x})\|_2^2\right] \\
&= (1 - \tau_2)\mathbb{E}_{\boldsymbol{x}\sim\mathcal{D}}\left[\tau_1(\boldsymbol{x})\|\nabla f_T(\boldsymbol{x})\|_2^2\right].
\end{aligned}
\tag{7}
$$

Plugging (7) into (6), we finally have

$$
\begin{aligned}
\|\nabla f_T - \nabla(g\circ f_S)\|_\mathcal{D}^2 \\
&= (1 - \tau_2)\mathbb{E}_{\boldsymbol{x}\sim\mathcal{D}}\left[\tau_1(\boldsymbol{x})\|\nabla f_T(\boldsymbol{x})\|_2^2\right] + \mathbb{E}_{\boldsymbol{x}\sim\mathcal{D}}(1 - \tau_1(\boldsymbol{x}))\|\nabla f_T(\boldsymbol{x})\|_2^2 \\
&= \mathbb{E}_{\boldsymbol{x}\sim\mathcal{D}}\left[(1 - \tau_2\tau_1(\boldsymbol{x}))\|\nabla f_T(\boldsymbol{x})\|_2^2\right] \\
&\leq (1 - \tau_1\tau_2)L^2,
\end{aligned}
$$

which completes the proof. $\qquad\square$

## D   PROOF OF THEOREM 2

**Theorem 2** (Restated). *For two functions $f_S : \mathbb{R}^n \to \mathbb{R}^m$, and $f_T : \mathbb{R}^n \to \mathbb{R}^d$, there is an affine function $g : \mathbb{R}^m \to \mathbb{R}^d$, so that*

$$\|\nabla f_T - \nabla(g \circ f_S)\|_{\mathcal{D}}^2 \leq 5\mathbb{E}_{\boldsymbol{x} \sim \mathcal{D}} \left( \begin{array}{l} \left((1 - \tau_1(\boldsymbol{x})\tau_2) + (1 - \tau_1(\boldsymbol{x}))(1 - \tau_2)\lambda_{f_T}(\boldsymbol{x})^2\right) \|\nabla f_T(\boldsymbol{x})\|_2^2 \\ + (\lambda_{f_T}(\boldsymbol{x}) + \lambda_{f_S}(\boldsymbol{x}))^2 \dfrac{\|\nabla f_S(\boldsymbol{x})\|_2^2}{\|\nabla f_S\|_{\mathcal{D},2}^2} \|\nabla f_T\|_{\mathcal{D},2}^2 \end{array} \right),$$

*where $g$ is defined as $g(\boldsymbol{z}) = \boldsymbol{A}\boldsymbol{z} + \boldsymbol{Const}$.*

*Moreover, if assuming that $f_T$ is L-Lipschitz continuous, i.e., $\|\nabla f_T(\boldsymbol{x})\|_2 \leq L$ for $\forall \boldsymbol{x} \sim supp(\mathcal{D})$, and considering the worst-case singular value ratio $\lambda$, we can have a more elegant statement:*

$$\|\nabla f_T - \nabla(g \circ f_S)\|_{\mathcal{D}}^2 \leq \left((1 - \tau_1\tau_2) + (1 - \tau_1)(1 - \tau_2)\lambda_{f_T}^2 + (\lambda_{f_T} + \lambda_{f_S})^2\right) 5L^2.$$

*Proof.* Recall that the matrix $\boldsymbol{A}$ is defined in Definition 3, i.e.,

$$\boldsymbol{A} = \text{proj}(\mathbb{E}_{\boldsymbol{x} \sim \mathcal{D}}[\Delta_{f_T, \boldsymbol{\delta}_{f_S}}(\boldsymbol{x})\Delta_{f_S, \boldsymbol{\delta}_{f_S}}(\boldsymbol{x})^\top] \left(\mathbb{E}_{\boldsymbol{x} \sim \mathcal{D}}[\Delta_{f_S, \boldsymbol{\delta}_{f_S}}(\boldsymbol{x})\Delta_{f_S, \boldsymbol{\delta}_{f_S}}(\boldsymbol{x})^\top]\right)^\dagger, \frac{\|\Delta_{f_T, \boldsymbol{\delta}_{f_S}}\|_{\mathcal{D}}}{\|\Delta_{f_S, \boldsymbol{\delta}_{f_S}}\|_{\mathcal{D}}}),$$

and we can see

$$\begin{aligned} \|\nabla f_T - \nabla(g \circ f_S)\|_{\mathcal{D},2}^2 &= \|\nabla f_T^\top - \nabla(g \circ f_S)^\top\|_{\mathcal{D},2}^2 = \|\nabla f_T^\top - \boldsymbol{A}\nabla f_S^\top\|_{\mathcal{D},2}^2 \\ &= \mathbb{E}_{\boldsymbol{x} \sim \mathcal{D}}\|\nabla f_T(\boldsymbol{x})^\top - \boldsymbol{A}\nabla f_S(\boldsymbol{x})^\top\|_2^2 \\ &= \mathbb{E}_{\boldsymbol{x} \sim \mathcal{D}} \max_{\|\boldsymbol{t}\|_2=1} \|\nabla f_T(\boldsymbol{x})^\top \boldsymbol{t} - \boldsymbol{A}\nabla f_S(\boldsymbol{x})^\top \boldsymbol{t}\|_2^2, \end{aligned} \tag{8}$$

where the last equality is due to the definition of matrix spectral norm.

Denoting $\nabla f^\top$ as either the Jacobian matrix $\nabla f_T^\top$ or $\nabla f_S^\top$, Singular Value Decomposition (SVD) suggests that $\nabla f(\boldsymbol{x})^\top = \boldsymbol{U}\boldsymbol{\Sigma}\boldsymbol{V}^\top$, where $\boldsymbol{\Sigma}$ is a diagonal matrix containing all singular values ordered by their absolute values. Let $\sigma_1, \cdots, \sigma_n$ denote ordered singular values. Nothing that the number of singular values that are non-zero may be less than $n$, so we fill the empty with zeros, such that each of them have corresponding singular vectors, i.e., the column vectors $\boldsymbol{v}_1, \cdots, \boldsymbol{v}_n$ in $\boldsymbol{V}$. That is being said, $\forall i \in [n]$, we have

$$\|\nabla f(\boldsymbol{x})^\top \boldsymbol{v}_i\|_2 = |\sigma_i|.$$

Let $\theta_i$ and $\boldsymbol{v}_i$ denote the singular values and vectors for $\nabla f_S(\boldsymbol{x})^\top$. Noting that $\{\boldsymbol{v}_i\}_{i=1}^n$ define a orthonormal basis for $\mathbb{R}^n$, we can represent

$$\boldsymbol{t} = \sum_{i=1}^n \theta_i \boldsymbol{v}_i, \tag{9}$$

where $\sum_{i=1}^n \theta_i^2 = 1$.

As adversarial attack is about the largest eigenvalue of the gradient, plugging (9) into (8), we can split it into two parts, i.e.,

$$\begin{aligned} (8) &= \mathbb{E}_{\boldsymbol{x} \sim \mathcal{D}} \max_{\|\boldsymbol{t}\|_2=1} \left\|\nabla f_T(\boldsymbol{x})^\top \left(\sum_{i=1}^n \theta_i \boldsymbol{v}_i\right) - \boldsymbol{A}\nabla f_S(\boldsymbol{x})^\top \left(\sum_{i=1}^n \theta_i \boldsymbol{v}_i\right)\right\|_2^2 \\ &= \mathbb{E}_{\boldsymbol{x} \sim \mathcal{D}} \max_{\|\boldsymbol{t}\|_2=1} \left\| \begin{array}{l} \nabla f_T(\boldsymbol{x})^\top (\theta_1 \boldsymbol{v}_1) - \boldsymbol{A}\nabla f_S(\boldsymbol{x})^\top (\theta_1 \boldsymbol{v}_1) \\ + \nabla f_T(\boldsymbol{x})^\top \left(\sum_{i=2}^n \theta_i \boldsymbol{v}_i\right) - \boldsymbol{A}\nabla f_S(\boldsymbol{x})^\top \left(\sum_{i=2}^n \theta_i \boldsymbol{v}_i\right) \end{array} \right\|_2^2. \end{aligned} \tag{10}$$

Denoting $\boldsymbol{u} = \sum_{i=2}^n \theta_i \boldsymbol{v}_i$, we can see this vector is orthogonal to $\boldsymbol{v}_1$. Let us denote $\boldsymbol{v}_1'$ as the singular vector with the biggest absolute singular value of $\nabla f_T(\boldsymbol{x})^\top$, parallel with attack $\boldsymbol{\delta}_{f_T}$. Now we split

$\boldsymbol{u} = \boldsymbol{u}_1 + \boldsymbol{u}_2$ into two terms, where $\boldsymbol{u}_1$ is parallel to $\boldsymbol{v}_1'$, and $\boldsymbol{u}_2$ is orthogonal to $\boldsymbol{u}_1$. As $\boldsymbol{u}_1$ is in the orthogonal space to $\boldsymbol{v}_1$ while parallel with $\boldsymbol{v}_1'$, it is bounded by the sine value of the angle between $\boldsymbol{v}_1$ and $\boldsymbol{v}_1'$, i.e., $\sqrt{1 - \tau_1(\boldsymbol{x})}$. Hence, noting that $\boldsymbol{u}$ is part of the unit vector $\boldsymbol{t}$,

$$\|\boldsymbol{u}_1\|_2 \leq \sqrt{1 - \tau_1(\boldsymbol{x})}\|\boldsymbol{u}\|_2 \leq \sqrt{1 - \tau_1(\boldsymbol{x})}. \tag{11}$$

Plugging $\boldsymbol{u}$ in (10), we have

$$(10) = \mathbb{E}_{\boldsymbol{x} \sim \mathcal{D}} \max_{\|\boldsymbol{t}\|_2=1} \left\| \begin{matrix} \nabla f_T(\boldsymbol{x})^\top (\theta_1 \boldsymbol{v}_1) - \boldsymbol{A} \nabla f_S(\boldsymbol{x})^\top (\theta_1 \boldsymbol{v}_1) \\ + \nabla f_T(\boldsymbol{x})^\top (\boldsymbol{u}_1 + \boldsymbol{u}_2) - \boldsymbol{A} \nabla f_S(\boldsymbol{x})^\top \boldsymbol{u} \end{matrix} \right\|_2^2$$

$$\leq \mathbb{E}_{\boldsymbol{x} \sim \mathcal{D}} \max_{\|\boldsymbol{t}\|_2=1} \left( \underbrace{\left\| \nabla f_T(\boldsymbol{x})^\top (\theta_1 \boldsymbol{v}_1) - \boldsymbol{A} \nabla f_S(\boldsymbol{x})^\top (\theta_1 \boldsymbol{v}_1) \right\|_2}_{X_1} + \underbrace{\left\| \nabla f_T(\boldsymbol{x})^\top \boldsymbol{u}_1 \right\|_2}_{X_2} + \underbrace{\left\| \nabla f_T(\boldsymbol{x})^\top \boldsymbol{u}_2 - \boldsymbol{A} \nabla f_S(\boldsymbol{x})^\top \boldsymbol{u} \right\|_2}_{X_3} \right)^2, \tag{12}$$

where the inequality is due to triangle inequality.

There are three terms we have to deal with, i.e., $X_1$, $X_2$ and $X_3$. Regarding the first term, $\boldsymbol{v}_1$ in $X_1$ aligns with the attack $\boldsymbol{\delta}_{f_S}(\boldsymbol{x})$, which we have known through adversarial attack. The second term $X_2$ is trivially bounded by (11). Although adversarial attacks tell us nothing about $X_3$, it can be bounded by the second largest singular values.

Let us first deal with two easiest, i.e., $X_2$ and $X_3$. Applying (11) on $X_2$ directly, we have

$$X_2 = \|\nabla f_T(\boldsymbol{x})^\top\|_2 \cdot \|\boldsymbol{u}_1\|_2 \leq \sqrt{1 - \tau_1(\boldsymbol{x})}\|\nabla f_T(\boldsymbol{x})^\top\|_2.$$

For $X_3$, noting that $\boldsymbol{u}_2$ is orthogonal to $\boldsymbol{v}_1'$, and $\boldsymbol{u}$ is orthogonal to $\boldsymbol{v}_1$, we can see that $\boldsymbol{u}_2$ has no components of the largest absolute singular vector of $\nabla f_T(\boldsymbol{x})^\top$, and $\boldsymbol{u}$ has no components of the largest absolute singular vector of $\nabla f_T(\boldsymbol{x})^\top$. Therefore,

$$\begin{aligned} X_3 &\leq \left\| \nabla f_T(\boldsymbol{x})^\top \boldsymbol{u}_2 \right\|_2 + \left\| \boldsymbol{A} \nabla f_S(\boldsymbol{x})^\top \boldsymbol{u} \right\|_2 \\ &\leq \sigma_{f_T,2}(\boldsymbol{x}) \|\boldsymbol{u}_2\|_2 + \sigma_{f_S,2}(\boldsymbol{x}) \|\boldsymbol{A}\|_2 \|\boldsymbol{u}\|_2 \\ &= \lambda_{f_T}(\boldsymbol{x}) \left\| \nabla f_T(\boldsymbol{x})^\top \right\|_2 \|\boldsymbol{u}_2\|_2 + \lambda_{f_S}(\boldsymbol{x}) \left\| \nabla f_S(\boldsymbol{x})^\top \right\|_2 \|\boldsymbol{A}\|_2 \|\boldsymbol{u}\|_2 \\ &\leq \lambda_{f_T}(\boldsymbol{x}) \left\| \nabla f_T(\boldsymbol{x})^\top \right\|_2 + \lambda_{f_S}(\boldsymbol{x}) \left\| \nabla f_S(\boldsymbol{x})^\top \right\|_2 \|\boldsymbol{A}\|_2, \end{aligned}$$

where the first inequality is due to triangle inequality, the second inequity is done by the attributes of singular values, and the definition of matrix 2-norm. The equality is done simply by applying the definition of singular values ratio (Definition 4), and the third inequality is due to the fact that $\|\boldsymbol{u}_2\|_2 \leq \|\boldsymbol{u}\|_2 \leq 1$.

Before dealing with $X_1$, let us simplify (12) by relax the square of summed terms to sum of squared terms, as the following.

$$\begin{aligned} (12) &= \mathbb{E}_{\boldsymbol{x} \sim \mathcal{D}} \max_{\|\boldsymbol{t}\|_2=1} (X_1 + X_2 + X_3)^2 \\ &= \mathbb{E}_{\boldsymbol{x} \sim \mathcal{D}} \max_{\|\boldsymbol{t}\|_2=1} X_1^2 + X_2^2 + X_3^2 + 2X_1 X_2 + 2X_2 X_3 + 2X_1 X_3 \\ &\leq \mathbb{E}_{\boldsymbol{x} \sim \mathcal{D}} \max_{\|\boldsymbol{t}\|_2=1} X_1^2 + X_2^2 + X_3^2 + 2\max\{X_1^2, X_2^2\} + 2\max\{X_2^2, X_3^2\} + 2\max\{X_1^2, X_3^2\} \\ &\leq \mathbb{E}_{\boldsymbol{x} \sim \mathcal{D}} \max_{\|\boldsymbol{t}\|_2=1} X_1^2 + X_2^2 + X_3^2 + 2(X_1^2 + X_2^2) + 2(X_2^2 + X_3^2) + 2(X_1^2 + X_3^2) \\ &= \mathbb{E}_{\boldsymbol{x} \sim \mathcal{D}} \max_{\|\boldsymbol{t}\|_2=1} 5(X_1^2 + X_2^2 + X_3^2). \tag{13} \end{aligned}$$

We note that this relaxation is not necessary, but simply for the simplicity of the final results without breaking what our theory suggests.

Bring what we we have about $X_2$ and $X_3$, and noting that $\theta_1 \leq 1$ depends on $t$, we can drop the max operation by

$$
(13) = \mathbb{E}_{x \sim \mathcal{D}} \max_{\|t\|_2 = 1} 5(X_1^2 + X_2^2 + X_3^2)
$$

$$
= \mathbb{E}_{x \sim \mathcal{D}} \max_{\|t\|_2 = 1} 5(\left\| \nabla f_T(x)^\top (\theta_1 v_1) - A \nabla f_S(x)^\top (\theta_1 v_1) \right\|_2^2 + X_2^2 + X_3^2)
$$

$$
\leq 5 \mathbb{E}_{x \sim \mathcal{D}} \left( \begin{array}{l} \left\| \nabla f_T(x)^\top v_1 - A \nabla f_S(x)^\top v_1 \right\|_2^2 + (1 - \tau_1(x)) \left\| \nabla f_T(x) \right\|_2^2 \\ + \left( (\lambda_{f_T}(x) + \lambda_{f_S}(x)) \left\| \nabla f_S(x)^\top \right\|_2 \|A\|_2 \right)^2 \end{array} \right). \quad (14)
$$

Now, let us deal with the first term. As $v_1$ is a unit vector and is in fact the direction of $f_S(x)$'s adversarial attack, we can write $\delta_{f_S,\epsilon}(x) = \epsilon v_1$. Hence,

$$
\mathbb{E}_{x \sim \mathcal{D}} \left\| \nabla f_T(x)^\top v_1 - A \nabla f_S(x)^\top v_1 \right\|_2^2
$$

$$
= \mathbb{E}_{x \sim \mathcal{D}} \frac{1}{\epsilon^2} \left\| \nabla f_T(x)^\top \delta_{f_S,\epsilon}(x) - A \nabla f_S(x)^\top \delta_{f_S,\epsilon}(x) \right\|_2^2
$$

$$
= \mathbb{E}_{x \sim \mathcal{D}} \frac{1}{\epsilon^2} \left\| \Delta_{f_T, \delta_{f_S}}(x) - A \Delta_{f_S, \delta_{f_S}}(x) \right\|_2^2, \quad (15)
$$

where the last equality is derived by applying the definition of $\Delta(x)$, i.e., equation (1). Note that we omit the $\epsilon$ in $\delta_{f_S,\epsilon}$ for notation simplicity.

The matrix $A$ is deigned to minimize (15), as shown in the proof of Proposition 1. Expanding the term we have

$$
(15) = \frac{1}{\epsilon^2} \mathbb{E}_{x \sim \mathcal{D}} \left[ \left\| \Delta_{f_T, \delta_{f_S}}(x) \right\|_2^2 + \left\| A \Delta_{f_S, \delta_{f_S}}(x) \right\|_2^2 - 2 \langle \Delta_{f_T, \delta_{f_S}}(x), A \Delta_{f_S, \delta_{f_S}}(x) \rangle \right]
$$

$$
= \frac{1}{\epsilon^2} \left( \left\| \Delta_{f_T, \delta_{f_S}} \right\|_{\mathcal{D}}^2 + \left\| A \Delta_{f_S, \delta_{f_S}} \right\|_{\mathcal{D}}^2 - 2 \langle \Delta_{f_T, \delta_{f_S}}, A \Delta_{f_S, \delta_{f_S}} \rangle_{\mathcal{D}} \right)
$$

$$
= \frac{\left\| \Delta_{f_T, \delta_{f_S}} \right\|_{\mathcal{D}}^2}{\epsilon^2} (1 - \tau_2)
$$

$$
= (1 - \tau_2) \mathbb{E}_{x \sim \mathcal{D}} \left\| \nabla f_T(x)^\top v_1 \right\|_2^2. \quad (16)
$$

Recall that $v_1$ is a unit vector aligns the direction of $\delta_{f_S}$, and we have used $v_1'$ to denote a unit vector that aligns the direction of $\delta_{f_T}$. As $\tau_1$ tells us about the angle between the two, let us split $v_1$ into to orthogonal vectors, i.e., $v_1 = \sqrt{\tau_1(x)} v_1' + \sqrt{1 - \tau_1(x)} v_{1,\perp}'$, where $v_{1,\perp}'$ is a unit vector that is orthogonal to $v_1'$.

Plugging this into (16) we have

$$
(16) = (1 - \tau_2) \mathbb{E}_{x \sim \mathcal{D}} \left\| \nabla f_T(x)^\top (\sqrt{\tau_1(x)} v_1' + \sqrt{1 - \tau_1(x)} v_{1,\perp}') \right\|_2^2
$$

$$
= (1 - \tau_2) \mathbb{E}_{x \sim \mathcal{D}} \left[ \left\| \nabla f_T(x)^\top \sqrt{\tau_1(x)} v_1' \right\|_2^2 + \left\| \nabla f_T(x)^\top \sqrt{1 - \tau_1(x)} v_{1,\perp}' \right\|_2^2 \right]
$$

$$
= (1 - \tau_2) \mathbb{E}_{x \sim \mathcal{D}} \left[ \tau_1(x) \left\| \nabla f_T(x)^\top \right\|_2^2 + (1 - \tau_1(x)) \lambda_{f_T}(x)^2 \left\| \nabla f_T(x)^\top \right\|_2^2 \right],
$$

where the second equality is due to the image of $v_1'$ and $v_{1,\perp}'$ after linear transformation $\nabla f_T(x)^\top$ are orthogonal, which can be easily observed through SVD.

Plugging this in (14), and with some regular algebra manipulation, finally we have

$$(14) = 5\mathbb{E}_{\boldsymbol{x}\sim\mathcal{D}}\begin{pmatrix} (1-\tau_2)\left[\tau_1(\boldsymbol{x})\left\|\nabla f_T(\boldsymbol{x})^\top\right\|_2^2 + (1-\tau_1(\boldsymbol{x}))\lambda_{f_T}(\boldsymbol{x})^2\left\|\nabla f_T(\boldsymbol{x})^\top\right\|_2^2\right] \\ +(1-\tau_1(\boldsymbol{x}))\left\|\nabla f_T(\boldsymbol{x})\right\|_2^2 \\ +(\lambda_{f_T}(\boldsymbol{x})+\lambda_{f_S}(\boldsymbol{x}))^2\left\|\nabla f_S(\boldsymbol{x})^\top\right\|_2^2\|\boldsymbol{A}\|_2^2 \end{pmatrix}$$

$$= 5\mathbb{E}_{\boldsymbol{x}\sim\mathcal{D}}\begin{pmatrix} (1-\tau_1(\boldsymbol{x})\tau_2)\left\|\nabla f_T(\boldsymbol{x})^\top\right\|_2^2 \\ +(1-\tau_1(\boldsymbol{x}))(1-\tau_2)\lambda_{f_T}(\boldsymbol{x})^2\left\|\nabla f_T(\boldsymbol{x})^\top\right\|_2^2 \\ +(\lambda_{f_T}(\boldsymbol{x})+\lambda_{f_S}(\boldsymbol{x}))^2\left\|\nabla f_S(\boldsymbol{x})^\top\right\|_2^2\|\boldsymbol{A}\|_2^2 \end{pmatrix}. \tag{17}$$

Recall that $\boldsymbol{A}$ is from a norm-restricted matrix space, i.e., the $\boldsymbol{A}$ is scaled so that its spectral norm is no greater than $\frac{\|\Delta_{f_T,\delta_{f_S}}\|_{\mathcal{D}}}{\|\Delta_{f_S,\delta_{f_S}}\|_{\mathcal{D}}}$, thus

$$\|\boldsymbol{A}\|_2^2 \le \frac{\|\Delta_{f_T,\delta_{f_S}}\|_{\mathcal{D}}^2}{\|\Delta_{f_S,\delta_{f_S}}\|_{\mathcal{D}}^2} \le \frac{\|\Delta_{f_T,\delta_{f_T}}\|_{\mathcal{D}}^2}{\|\Delta_{f_S,\delta_{f_S}}\|_{\mathcal{D}}^2}$$

$$= \frac{\mathbb{E}_{\boldsymbol{x}\sim\mathcal{D}}\|\Delta_{f_T,\delta_{f_T}}(\boldsymbol{x})\|_2^2}{\mathbb{E}_{\boldsymbol{x}\sim\mathcal{D}}\|\Delta_{f_S,\delta_{f_S}}(\boldsymbol{x})\|_2^2} = \frac{\mathbb{E}_{\boldsymbol{x}\sim\mathcal{D}}\|\nabla f_T^\top(\boldsymbol{x})\|_2^2}{\mathbb{E}_{\boldsymbol{x}\sim\mathcal{D}}\|\nabla f_S^\top(\boldsymbol{x})\|_2^2}$$

$$= \frac{\|\nabla f_T^\top\|_{\mathcal{D},2}^2}{\|\nabla f_S^\top\|_{\mathcal{D},2}^2}. \tag{18}$$

Hence, plugging the above inequality to (17), the first statement of the theorem is proven, i.e.,

$$(17) \le 5\mathbb{E}_{\boldsymbol{x}\sim\mathcal{D}}\begin{pmatrix} (1-\tau_1(\boldsymbol{x})\tau_2)\left\|\nabla f_T(\boldsymbol{x})^\top\right\|_2^2 \\ +(1-\tau_1(\boldsymbol{x}))(1-\tau_2)\lambda_{f_T}^2\left\|\nabla f_T(\boldsymbol{x})^\top\right\|_2^2 \\ +(\lambda_{f_T}(\boldsymbol{x})+\lambda_{f_S}(\boldsymbol{x}))^2\left\|\nabla f_S(\boldsymbol{x})^\top\right\|_2^2\frac{\|\nabla f_T^\top\|_{\mathcal{D},2}^2}{\|\nabla f_S^\top\|_{\mathcal{D},2}^2} \end{pmatrix}. \tag{19}$$

To see the second statement of the theorem, we assume $f_T$ is $L$-Lipschitz continuous, i.e., $\|\nabla f_T(\boldsymbol{x})\|_2 \le L$ for $\forall \boldsymbol{x} \in \text{supp}(\mathcal{D})$, and considering the worst-case singular value ratio $\lambda = \max_{\boldsymbol{x}\in\text{supp}(\mathcal{D})}$ for either $f_S, f_T$, we can continue as

$$(19) \le 5\begin{pmatrix} \mathbb{E}_{\boldsymbol{x}\sim\mathcal{D}}\left[(1-\tau_1(\boldsymbol{x})\tau_2)\left\|\nabla f_T(\boldsymbol{x})^\top\right\|_2^2\right] \\ +\mathbb{E}_{\boldsymbol{x}\sim\mathcal{D}}\left[(1-\tau_1(\boldsymbol{x}))(1-\tau_2)\lambda_{f_T}^2\left\|\nabla f_T(\boldsymbol{x})^\top\right\|_2^2\right] \\ +\mathbb{E}_{\boldsymbol{x}\sim\mathcal{D}}\left[(\lambda_{f_T}+\lambda_{f_S})^2\left\|\nabla f_S(\boldsymbol{x})^\top\right\|_2^2\frac{\|\nabla f_T^\top\|_{\mathcal{D},2}^2}{\|\nabla f_S^\top\|_{\mathcal{D},2}^2}\right] \end{pmatrix}$$

$$= 5\begin{pmatrix} \mathbb{E}_{\boldsymbol{x}\sim\mathcal{D}}\left[(1-\tau_1(\boldsymbol{x})\tau_2)\left\|\nabla f_T(\boldsymbol{x})^\top\right\|_2^2\right] \\ +\mathbb{E}_{\boldsymbol{x}\sim\mathcal{D}}\left[(1-\tau_1(\boldsymbol{x}))(1-\tau_2)\lambda_{f_T}^2\left\|\nabla f_T(\boldsymbol{x})^\top\right\|_2^2\right] \\ +(\lambda_{f_T}+\lambda_{f_S})^2\|\nabla f_T^\top\|_{\mathcal{D},2}^2 \end{pmatrix}$$

$$= 5\mathbb{E}_{\boldsymbol{x}\sim\mathcal{D}}\left((1-\tau_1(\boldsymbol{x})\tau_2)+(1-\tau_1(\boldsymbol{x}))(1-\tau_2)\lambda_{f_T}^2+(\lambda_{f_T}+\lambda_{f_S})^2\right)\left\|\nabla f_T(\boldsymbol{x})^\top\right\|_2^2$$

$$\le \mathbb{E}_{\boldsymbol{x}\sim\mathcal{D}}\left((1-\tau_1(\boldsymbol{x})\tau_2)+(1-\tau_1(\boldsymbol{x}))(1-\tau_2)\lambda_{f_T}^2+(\lambda_{f_T}+\lambda_{f_S})^2\right)5L^2$$

$$= \left((1-\tau_1\tau_2)+(1-\tau_1)(1-\tau_2)\lambda_{f_T}^2+(\lambda_{f_T}+\lambda_{f_S})^2\right)5L^2,$$

where the first inequality is due to the definition of worst-case singular value ratio, the last inequality is by Lipschitz condition, and the last equality is done be simply applying the definition of $\tau_1$.

$\square$

# E  PROOF OF THEOREM 3

The idea for proving Theorem 3 is straight-forward: bounded gradients difference implies bounded function difference, and then bounded function difference implies bounded loss difference.

To begin with, let us prove the following lemma.

**Lemma 1.** *Without loss of generality we assume $\|\boldsymbol{x}\|_2 \le 1$ for $\forall \boldsymbol{x} \in supp(\mathcal{D})$. Consider functions $f_S : \mathbb{R}^n \to \mathbb{R}^m$, $f_T : \mathbb{R}^n \to \mathbb{R}^d$, and an affine function $g : \mathbb{R}^m \to \mathbb{R}^d$, suggested by Theorem 1 or Theorem 2, such that $g(f_S(\boldsymbol{0})) = f_T(\boldsymbol{0})$, if both $f_T, f_S$ are $\beta$-smooth in $\{\boldsymbol{x} \mid \|\boldsymbol{x}\| \le 1\}$, we have*

$$\|f_T - g \circ f_S\|_{\mathcal{D}} \le \|\nabla f_T - \nabla(g \circ f_S)\|_{\mathcal{D},2} + \left(1 + \frac{\|\nabla f_T\|_{\mathcal{D},2}}{\|\nabla f_S\|_{\mathcal{D},2}}\right)\beta.$$

*Proof.* Let us denote $v(\boldsymbol{x}) = f_T(\boldsymbol{x}) - g \circ f_S(\boldsymbol{x})$, and we can show the smoothness of $v(\cdot)$.

As $g(\cdot)$ is an affine function satisfying $g(f_S(\boldsymbol{0})) = f_T(\boldsymbol{0})$, it can be denoted as $g(\boldsymbol{z}) = \boldsymbol{A}(\boldsymbol{z} - f_S(\boldsymbol{0})) + f_T(\boldsymbol{0})$, where $\boldsymbol{A}$ is a matrix suggested by Theorem 1 or Theorem 2. Therefore, denoting $\mathbb{B}_1 = \{\boldsymbol{x} \mid \|\boldsymbol{x}\| \le 1\}$ as a unit ball, for $\forall \boldsymbol{x}, \boldsymbol{y} \in \mathbb{B}_1$ it holds that

$$
\begin{aligned}
\|\nabla v(\boldsymbol{x}) - \nabla v(\boldsymbol{y})\|_2 &= \left\|\nabla v(\boldsymbol{x})^\top - \nabla v(\boldsymbol{y})^\top\right\|_2 \\
&= \left\|\nabla f_T(\boldsymbol{x})^\top - \nabla f_T(\boldsymbol{y})^\top - \boldsymbol{A}(\nabla f_S(\boldsymbol{x})^\top - \nabla f_S(\boldsymbol{y})^\top)\right\|_2 \\
&\le \left\|\nabla f_T(\boldsymbol{x})^\top - \nabla f_T(\boldsymbol{y})^\top\right\|_2 + \left\|\boldsymbol{A}(\nabla f_S(\boldsymbol{x})^\top - \nabla f_S(\boldsymbol{y})^\top)\right\|_2 \\
&\le \left\|\nabla f_T(\boldsymbol{x})^\top - \nabla f_T(\boldsymbol{y})^\top\right\|_2 + \|\boldsymbol{A}\|_2 \left\|\nabla f_S(\boldsymbol{x})^\top - \nabla f_S(\boldsymbol{y})^\top\right\|_2, \quad (20)
\end{aligned}
$$

where the last second inequality is due to triangle inequality, and the last inequality is by the property of spectral norm.

Applying the $\beta$-smoothness of $f_S$ and $f_T$, and noting that $\|\boldsymbol{A}\|_2 \le \frac{\|\nabla f_T\|_{\mathcal{D},2}}{\|\nabla f_S\|_{\mathcal{D},2}}$ as shown in (18), we can continue as

$$(20) \le \beta\|\boldsymbol{x} - \boldsymbol{y}\|_2 + \|\boldsymbol{A}\|_2 \beta\|\boldsymbol{x} - \boldsymbol{y}\|_2 \le \beta\|\boldsymbol{x} - \boldsymbol{y}\|_2 + \frac{\|\nabla f_T\|_{\mathcal{D},2}}{\|\nabla f_S\|_{\mathcal{D},2}}\beta\|\boldsymbol{x} - \boldsymbol{y}\|_2$$

$$= \left(1 + \frac{\|\nabla f_T\|_{\mathcal{D},2}}{\|\nabla f_S\|_{\mathcal{D},2}}\right)\beta\|\boldsymbol{x} - \boldsymbol{y}\|_2,$$

which suggests that $v(\cdot)$ is $\left(1 + \frac{\|\nabla f_T\|_{\mathcal{D},2}}{\|\nabla f_S\|_{\mathcal{D},2}}\right)\beta$-smooth.

We are ready to prove the lemma now. Applying the mean value theorem, for $\forall \boldsymbol{x} \in \mathbb{B}_1$, we have

$$v(\boldsymbol{x}) - v(\boldsymbol{0}) = \nabla v(\xi\boldsymbol{x})^\top \boldsymbol{x},$$

where $\xi \in (0, 1)$ is a scalar number. Subtracting $\nabla v(\boldsymbol{x})^\top \boldsymbol{x}$ on both sides give

$$v(\boldsymbol{x}) - v(\boldsymbol{0}) - \nabla v(\boldsymbol{x})^\top \boldsymbol{x} = (\nabla v(\xi\boldsymbol{x}) - \nabla v(\boldsymbol{x}))^\top \boldsymbol{x}$$

$$\left\|v(\boldsymbol{x}) - v(\boldsymbol{0}) - \nabla v(\boldsymbol{x})^\top \boldsymbol{x}\right\|_2 = \left\|(\nabla v(\xi\boldsymbol{x}) - \nabla v(\boldsymbol{x}))^\top \boldsymbol{x}\right\|_2$$

$$\left\|v(\boldsymbol{x}) - v(\boldsymbol{0}) - \nabla v(\boldsymbol{x})^\top \boldsymbol{x}\right\|_2 \le \|(\nabla v(\xi\boldsymbol{x}) - \nabla v(\boldsymbol{x}))\|_2 \|\boldsymbol{x}\|_2.$$

Let us denote $\beta_1 = \left(1 + \frac{\|\nabla f_T\|_{\mathcal{D},2}}{\|\nabla f_S\|_{\mathcal{D},2}}\right)\beta$ for notation convenience, and apply the definition of smoothness:

$$\|v(\boldsymbol{x}) - v(\boldsymbol{0}) - \nabla v(\boldsymbol{x})^\top \boldsymbol{x}\|_2 \le \beta_1(1 - \xi)\|\boldsymbol{x}\|_2^2 \le \beta_1. \quad (21)$$

Noting that $v(\boldsymbol{0}) = 0$ and applying the triangle inequality, we have

$$\|v(\boldsymbol{x}) - v(\boldsymbol{0}) - \nabla v(\boldsymbol{x})^\top \boldsymbol{x}\|_2 \ge \|v(\boldsymbol{x})\|_2 - \|\nabla v(\boldsymbol{x})^\top \boldsymbol{x}\|_2 \ge \|v(\boldsymbol{x})\|_2 - \|\nabla v(\boldsymbol{x})^\top\|_2$$

Plugging it into (21), we have

$$\|v(\boldsymbol{x})\|_2 \le \beta_1 + \|\nabla v(\boldsymbol{x})^\top\|_2$$

$$\|v(\boldsymbol{x})\|_2^2 \le \beta_1^2 + \|\nabla v(\boldsymbol{x})^\top\|_2^2 + 2\beta_1\|\nabla v(\boldsymbol{x})^\top\|_2$$

$$\mathbb{E}_{\boldsymbol{x} \sim \mathcal{D}}\|v(\boldsymbol{x})\|_2^2 \le \beta_1^2 + \mathbb{E}_{\boldsymbol{x} \sim \mathcal{D}}\|\nabla v(\boldsymbol{x})^\top\|_2^2 + 2\beta_1\mathbb{E}_{\boldsymbol{x} \sim \mathcal{D}}\|\nabla v(\boldsymbol{x})^\top\|_2$$

$$\mathbb{E}_{\boldsymbol{x} \sim \mathcal{D}}\|v(\boldsymbol{x})\|_2^2 \le \beta_1^2 + \mathbb{E}_{\boldsymbol{x} \sim \mathcal{D}}\|\nabla v(\boldsymbol{x})\|_2^2 + 2\beta_1\mathbb{E}_{\boldsymbol{x} \sim \mathcal{D}}\|\nabla v(\boldsymbol{x})\|_2$$

$$\|v\|_{\mathcal{D}}^2 \le \beta_1^2 + \|\nabla v\|_{\mathcal{D},2}^2 + 2\beta_1\mathbb{E}_{\boldsymbol{x} \sim \mathcal{D}}\|\nabla v(\boldsymbol{x})\|_2$$

Applying Jensen's inequality to the last term, we get

$$
\begin{aligned}
\|v\|_{\mathcal{D}}^2 &\leq \beta_1^2 + \|\nabla v\|_{\mathcal{D},2}^2 + 2\beta_1\sqrt{\mathbb{E}_{\boldsymbol{x}\sim\mathcal{D}}\|\nabla v(\boldsymbol{x})\|_2^2} \\
&= \beta_1^2 + \|\nabla v\|_{\mathcal{D},2}^2 + 2\beta_1\sqrt{\|\nabla v\|_{\mathcal{D},2}^2} = \beta_1^2 + \|\nabla v\|_{\mathcal{D},2}^2 + 2\beta_1\|\nabla v\|_{\mathcal{D},2} \\
&= (\|\nabla v\|_{\mathcal{D},2} + \beta_1)^2
\end{aligned}
$$

Plugging $\beta_1 = \left(1 + \frac{\|\nabla f_T\|_{\mathcal{D},2}}{\|\nabla f_S\|_{\mathcal{D},2}}\right)\beta$ and $v = f_T - g \circ f_S$ into the above inequality completes the proof. $\qquad\square$

With the above lemma, it is easy to show the mean squared loss on the transferred model is also bounded.

**Theorem 3** (Restated). *Without loss of generality we assume $\|\boldsymbol{x}\|_2 \leq 1$ for $\forall \boldsymbol{x} \in supp(\mathcal{D})$. Consider functions $f_S : \mathbb{R}^n \to \mathbb{R}^m$, $f_T : \mathbb{R}^n \to \mathbb{R}^d$, and an affine function $g : \mathbb{R}^m \to \mathbb{R}^d$, suggested by Theorem 1 or Theorem 2, such that $g(f_S(\boldsymbol{0})) = f_T(\boldsymbol{0})$. If both $f_T, f_S$ are $\beta$-smooth, then*

$$
\|g \circ f_S - y\|_{\mathcal{D}}^2 \leq \left(\|f_T - y\|_{\mathcal{D}} + \|\nabla f_T - \nabla g \circ f_S\|_{\mathcal{D},2} + \left(1 + \frac{\|\nabla f_T\|_{\mathcal{D},2}}{\|\nabla f_S\|_{\mathcal{D},2}}\right)\beta\right)^2
$$

*Proof.* Let us denote $\beta_1 = \left(1 + \frac{\|\nabla f_T\|_{\mathcal{D},2}}{\|\nabla f_S\|_{\mathcal{D},2}}\right)\beta$, and according to Lemma 1 we can see

$$
\|f_T - g \circ f_S\|_{\mathcal{D}} \leq \|\nabla f_T - \nabla(g \circ f_S)\|_{\mathcal{D},2} + \beta_1 \tag{22}
$$

Applying a standard algebra manipulation to the left hand side, and then applying triangle inequality, we have

$$
\|f_T - g \circ f_S\|_{\mathcal{D}} = \|f_T - y + y - g \circ f_S\|_{\mathcal{D}} \geq \|y - g \circ f_S\|_{\mathcal{D}} - \|f_T - y\|_{\mathcal{D}}.
$$

Plugging this directly into (22), it holds that

$$
\begin{aligned}
\|y - g \circ f_S\|_{\mathcal{D}} - \|f_T - y\|_{\mathcal{D}} &\leq \|\nabla f_T - \nabla(g \circ f_S)\|_{\mathcal{D},2} + \beta_1 \\
\|y - g \circ f_S\|_{\mathcal{D}} &\leq \|f_T - y\|_{\mathcal{D}} + \|\nabla f_T - \nabla(g \circ f_S)\|_{\mathcal{D},2} + \beta_1
\end{aligned}
$$

Replacing $\beta_1$ by $\left(1 + \frac{\|\nabla f_T\|_{\mathcal{D},2}}{\|\nabla f_S\|_{\mathcal{D},2}}\right)\beta$ and taking the square, we can see Theorem 3 is proven. $\qquad\square$

## F  PROOF OF PROPOSITION 2

**Proposition 2** (Restated). *If $\ell_T$ is mean squared loss and $f_T$ achieves zero loss on $\mathcal{D}$, then the adversarial loss defined in Definition 6 is approximately upper and lower bounded by*

$$
\mathcal{L}_{adv}(f_T, \boldsymbol{\delta}_{f_S}; y, \mathcal{D}) \geq \epsilon^2 \mathbb{E}_{\boldsymbol{x}\sim\mathcal{D}}\left[\tau_1(\boldsymbol{x})\|\nabla f_T(\boldsymbol{x})\|_2^2\right] + O(\epsilon^3),
$$

$$
\mathcal{L}_{adv}(f_T, \boldsymbol{\delta}_{f_S}; y, \mathcal{D}) \leq \epsilon^2 \mathbb{E}_{\boldsymbol{x}\sim\mathcal{D}}\left[\left(\lambda_{f_T}^2 + (1 - \lambda_{f_T}^2)\tau_1(\boldsymbol{x})\right)\|\nabla f_T(\boldsymbol{x})\|_2^2\right] + O(\epsilon^3),
$$

*where $O(\epsilon^3)$ denotes a cubic error term.*

*Proof.* Recall that the empirical adversarial transferability is defined as a loss

$$
\mathcal{L}_{adv}(f_T, \boldsymbol{\delta}_{f_S,\epsilon}; y, \mathcal{D}) = \mathbb{E}_{\boldsymbol{x}\sim\mathcal{D}} \quad \ell_T(f_T(\boldsymbol{x} + \boldsymbol{\delta}_{f_S,\epsilon}(\boldsymbol{x})), y(\boldsymbol{x})).
$$

As $\ell_T$ is mean squared loss, and $f_T$ achieves zero loss, i.e., $f_T = y$, we have

$$
\begin{aligned}
\mathcal{L}_{adv}(f_T, \boldsymbol{\delta}_{f_S,\epsilon}; y, \mathcal{D}) &= \mathbb{E}_{\boldsymbol{x}\sim\mathcal{D}} \|f_T(\boldsymbol{x} + \boldsymbol{\delta}_{f_S,\epsilon}(\boldsymbol{x})) - y(\boldsymbol{x})\|_2^2 \\
&= \mathbb{E}_{\boldsymbol{x}\sim\mathcal{D}} \|f_T(\boldsymbol{x} + \boldsymbol{\delta}_{f_S,\epsilon}(\boldsymbol{x})) - f_T(\boldsymbol{x})\|_2^2.
\end{aligned}
$$

Denoting $\boldsymbol{\delta}_{f_S,\epsilon}(\boldsymbol{x}) = \epsilon\boldsymbol{\delta}_{f_S,1}(\boldsymbol{x})$, and define an auxiliary function $h$ as

$$
h(t) = f_T(\boldsymbol{x} + t\boldsymbol{\delta}_{f_S,1}(\boldsymbol{x})) - f_T(\boldsymbol{x}),
$$

we can see that $\|f_T(\boldsymbol{x} + \boldsymbol{\delta}_{f_S,\epsilon}(\boldsymbol{x})) - f_T(\boldsymbol{x})\|_2^2 = \|h(\epsilon)\|_2^2$.

We can then apply Taylor expansion to approximate $h(\epsilon)$ with a second order error term $O(\epsilon^2)$, i.e.,

$$h(\epsilon) = \frac{\partial h}{\partial t}\big|_{t=0} + O(\epsilon^2) = \epsilon \nabla f_T(\boldsymbol{x})^\top \boldsymbol{\delta}_{f_S,1} + O(\epsilon^2).$$

Therefore, assuming that $\|\nabla f_T(\boldsymbol{x})\|_2$ is bounded for $\boldsymbol{x} \in \text{supp}(\mathcal{D})$, we have

$$\|f_T(\boldsymbol{x} + \boldsymbol{\delta}_{f_S,\epsilon}(\boldsymbol{x})) - f_T(\boldsymbol{x})\|_2^2 = \|h(\epsilon)\|_2^2 = \epsilon^2 \left\|\nabla f_T(\boldsymbol{x})^\top \boldsymbol{\delta}_{f_S,1}(\boldsymbol{x})\right\|_2^2 + O(\epsilon^3), \qquad (23)$$

where we have omit higher order error term, i.e., $O(\epsilon^4)$.

Next, let us deal with the term $\left\|\nabla f_T(\boldsymbol{x})^\top \boldsymbol{\delta}_{f_S,1}(\boldsymbol{x})\right\|_2^2$. Same us the technique we use in the proof of Theorem 2, we split $\boldsymbol{\delta}_{f_S,1}(\boldsymbol{x}) = \boldsymbol{v}_1 + \boldsymbol{v}_2$, where $\boldsymbol{v}_1$ aligns the direction of $\boldsymbol{\delta}_{f_T,1}(\boldsymbol{x})$, and $\boldsymbol{v}_2$ is orthogonal to $\boldsymbol{v}_1$. Noting that $\tau_1(\boldsymbol{x})$ is the squared cosine of the angle between $\boldsymbol{\delta}_{f_S,1}(\boldsymbol{x})$ and $\boldsymbol{\delta}_{f_T,1}(\boldsymbol{x})$, we can see that

$$\|\boldsymbol{v}_1\|_2^2 = \tau_1(\boldsymbol{x}) \|\boldsymbol{\delta}_{f_S,1}(\boldsymbol{x})\|_2^2 = \tau_1(\boldsymbol{x}),$$
$$\|\boldsymbol{v}_2\|_2^2 = (1 - \tau_1(\boldsymbol{x})) \|\boldsymbol{\delta}_{f_S,1}(\boldsymbol{x})\|_2^2 = (1 - \tau_1(\boldsymbol{x})).$$

Therefore, we can continue as

$$\begin{aligned}
\left\|\nabla f_T(\boldsymbol{x})^\top \boldsymbol{\delta}_{f_S,1}(\boldsymbol{x})\right\|_2^2 &= \left\|\nabla f_T(\boldsymbol{x})^\top (\boldsymbol{v}_1 + \boldsymbol{v}_2)\right\|_2^2 \\
&= \left\|\nabla f_T(\boldsymbol{x})^\top \boldsymbol{v}_1\right\|_2^2 + \left\|\nabla f_T(\boldsymbol{x})^\top \boldsymbol{v}_2\right\|_2^2 \\
&= \tau_1(\boldsymbol{x}) \|\nabla f_T(\boldsymbol{x})\|_2^2 + \left\|\nabla f_T(\boldsymbol{x})^\top \boldsymbol{v}_2\right\|_2^2, \qquad (24)
\end{aligned}$$

where the second equality is because that $\boldsymbol{v}_1$ is corresponding to the largest singular value of $\nabla f_T(\boldsymbol{x})^\top$, and $\boldsymbol{v}_2$ is orthogonal to $\boldsymbol{v}_1$.

Next, we derive the lower bound and upper bound for (24). The lower bounded can be derived as

$$\tau_1(\boldsymbol{x}) \|\nabla f_T(\boldsymbol{x})\|_2^2 + \left\|\nabla f_T(\boldsymbol{x})^\top \boldsymbol{v}_2\right\|_2^2 \geq \tau_1(\boldsymbol{x}) \|\nabla f_T(\boldsymbol{x})\|_2^2,$$

and the upper bounded can be derived as

$$\begin{aligned}
\tau_1(\boldsymbol{x}) \|\nabla f_T(\boldsymbol{x})\|_2^2 + \left\|\nabla f_T(\boldsymbol{x})^\top \boldsymbol{v}_2\right\|_2^2 &\leq \tau_1(\boldsymbol{x}) \|\nabla f_T(\boldsymbol{x})\|_2^2 + \lambda_{f_T}(\boldsymbol{x})^2 \|\nabla f_T(\boldsymbol{x})\|_2^2 \|\boldsymbol{v}_2\|_2^2 \\
&= \tau_1(\boldsymbol{x}) \|\nabla f_T(\boldsymbol{x})\|_2^2 + \lambda_{f_T}(\boldsymbol{x})^2 \|\nabla f_T(\boldsymbol{x})\|_2^2 (1 - \tau_1(\boldsymbol{x})) \\
&\leq \tau_1(\boldsymbol{x}) \|\nabla f_T(\boldsymbol{x})\|_2^2 + \lambda_{f_T}^2 \|\nabla f_T(\boldsymbol{x})\|_2^2 (1 - \tau_1(\boldsymbol{x})) \\
&= \left(\lambda_{f_T}^2 + (1 - \lambda_{f_T}^2)\tau_1(\boldsymbol{x})\right) \|\nabla f_T(\boldsymbol{x})\|_2^2,
\end{aligned}$$

where $\lambda_{f_T}(\boldsymbol{x})$ is the singular value ratio of $f_T$ at $\boldsymbol{x}$, and $\lambda_{f_T}$ is the maximal singular value of $f_T$.

Applying the lower and upper bound to (23), we finally have

$$\begin{aligned}
\|f_T(\boldsymbol{x} + \boldsymbol{\delta}_{f_S,\epsilon}(\boldsymbol{x})) - f_T(\boldsymbol{x})\|_2^2 &\geq \epsilon^2 \tau_1(\boldsymbol{x}) \|\nabla f_T(\boldsymbol{x})\|_2^2 + O(\epsilon^3), \\
\|f_T(\boldsymbol{x} + \boldsymbol{\delta}_{f_S,\epsilon}(\boldsymbol{x})) - f_T(\boldsymbol{x})\|_2^2 &\leq \epsilon^2 \left(\lambda_{f_T}^2 + (1 - \lambda_{f_T}^2)\tau_1(\boldsymbol{x})\right) \|\nabla f_T(\boldsymbol{x})\|_2^2 + O(\epsilon^3). \qquad (25)
\end{aligned}$$

Noting that

$$\mathcal{L}_{adv}(f_T, \boldsymbol{\delta}_{f_S,\epsilon}; y, \mathcal{D}) = \mathbb{E}_{\boldsymbol{x} \sim \mathcal{D}} \|f_T(\boldsymbol{x} + \boldsymbol{\delta}_{f_S,\epsilon}(\boldsymbol{x})) - f_T(\boldsymbol{x})\|_2^2,$$

we can see that taking expectation to (25) completes the proof. $\qquad \square$

## G  EXPERIMENT DETAILS

All experiments are conducted on 4 RTX 2080 Ti GPUs and in python3 Ubuntu 16.04 environment.

## G.1 ATTACK METHODS

**PGD Attack** is generated iteratively: denote step size as $\xi$, the source model as $f_S$, and the loss function on the source problem. $\ell_S(\cdot, \cdot)$. We initialize $\boldsymbol{x}_0$ to be uniformly sampled from the $\epsilon$-ball $\mathbb{B}_\epsilon(\boldsymbol{x})$ of radius $\epsilon$ centered as instance $\boldsymbol{x}$, and then generate the adversarial instance iteratively: at step $t$ we compute $\boldsymbol{x}_{t+1} = \boldsymbol{x}_t + \xi \cdot \mathbf{sign}(\nabla_{\boldsymbol{x}_t} \ell_S(f_S(\boldsymbol{x}_t), f_S(\boldsymbol{x})))$. Denoting the adversarial example at instance $\boldsymbol{x}$ using PGD on source model $f_S$ as $PGD_{f_S}(\boldsymbol{x})$, we measure the *adversarial loss* from $f_S$ to $f_T$ based on the loss $\ell_T(\cdot, y)$ of $f_T$ on target data $\mathcal{D}$ given attacks generated on $f_S$, i.e.,

$$\mathcal{L}_T(f_T \circ PGD_{f_S}; y, \mathcal{D}) \quad = \quad \mathbb{E}_{\boldsymbol{x} \sim \mathcal{D}} \quad \ell_T(f_T(PGD_{f_S}(\boldsymbol{x})), y(\boldsymbol{x})).$$

**TextFooler** iteratively replaces words in target sentences by looking up similar words in the dictionary. It pauses when the predicted label is changed or runs out of the attack budget. We modify it such that it pauses when the percentage of changed words reaches 10%.

## G.2 ADVERSARIAL TRANSFERABILITY INDICATES KNOWLEDGE-TRANSFER AMONG DATA DISTRIBUTIONS

**Details of Dataset construction** For the image domain, we divide the classes of the original datasets into two categories, animals (bird, cat, deer, dog) and transportation vehicles (airplane, automobile, ship, truck). Each of the source datasets consists of different a percentage of animals and transportation vehicles, while the target dataset contains only transportation vehicles, which is meant to control the closeness of the two data distributions.

**Details of Model Training** Image: we train five source models on the five source datasets from $0\%$ animals to $100\%$ animals, and one reference models on STL-10 with identical architectures and hyperparameters. We use SGD optimizer and standard cross-entropy loss with learning rate 0.1, momentum 0.9, and weight decay $10^{-4}$. Each model is trained for 300 epochs.
Natural Language: we fine-tune a Bert on each of the datasets with Adam and learning rate 0.0003 for 100 epochs. For transferred models, we run Adam with a smaller learning rate 0.0001 for 3 epochs.

## G.3 ADVERSARIAL TRANSFERABILITY INDICATING KNOWLEDGE-TRANSFER AMONG ATTRIBUTES

**Details of Model Training** We train 40 binary source classifiers on each of the 40 attributes of CelebA with ResNet18 (He et al., 2016). All the classifiers are trained with optimizer Adadelta with a learning rate of 1.0 for 14 epochs. We also train a facial recognition model as a reference model on CelebA with 10,177 identities using ResNet18 as the controlled experiment.The reference facial recognition model is optimized with SGD and initial learning rate 0.1 on the ArcFace (Deng et al., 2019) with focal loss (Lin et al., 2017) for 125 epochs. For each source model, we construct a transferred model by stripping off the last layers and attaching a facial recognition head without parameters. Then we use the 40 transferred models to evaluate the knowledge transferability on 7 facial recognition benchmarks.

## G.4 ADVERSARIAL TRANSFERABILITY INDICATING KNOWLEDGE-TRANSFER AMONG TASKS

**Details of Model Training** We use 15 pretrained models released in the task bank (Zamir et al., 2018) as the source models. Each source model consists of two parts, an encoder, and a decoder. The encoder is a modified ResNet50 without pooling, homogeneous across all tasks, whereas the decoder is customized to suit the output of each task. When measuring the adversarial transferability, we will use each source model as a reference model and compute the transferability matrix as described below.

*Adversarial Transferability Matrix (ATM)* is used here to measure the adversarial transferability between multiple tasks, modified from the *Affinity Matrix* in (Zamir et al., 2018). In the experiment of determining similarity among tasks, it is hard to compare directly and fairly, since each task is of different loss functions, which is usually in a very different scale with each other. To solve this problem, we take the same ordinal normalization approach as Zamir et al. (2018). Suppose we have N tasks in the pool, a tournament matrix $M_T$ for each task T is constructed, where the element of the matrix $m_{i,j}$ represents what percentages of adversarial examples generated from the $i$th task transfers better to task T than the ones of the $j$th task (untargeted attack success rate is used here).

Then we take the principal eigenvectors of the N tournament matrices and stack them together to build the $N \times N$ adversarial transferability matrix. To generate the corresponding "task categories" for comparison, we sample 1000 images from the public dataset and perform a virtual adversarial attack on each of the 15 source models. Adversarial perturbation with $\epsilon$ ($L_\infty$ norm) as 0.03,0.06 are used and we run 10 steps PGD-based attack for efficiency. Then we measure these adversarial examples' effectiveness on each of the 15 tasks by the corresponding loss functions. After we obtain the $15 \times 15$ ATM, we take columns of this matrix as features for each task and perform agglomerative clustering to obtain the Task Similarity Tree.

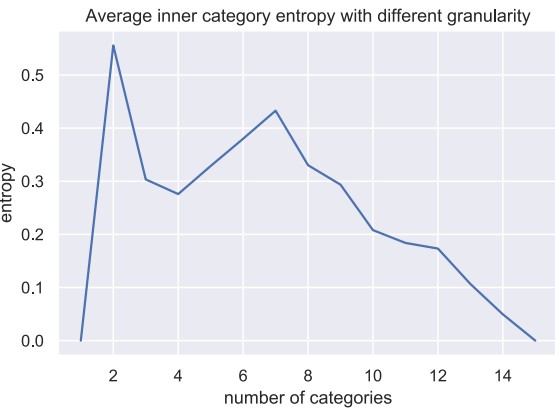

Figure 5: We also quantitatively compare our prediction with the Taskonomy (Zamir et al., 2018) prediction when different number of categories is enforced. We find our prediction is similar with theirs with $n \geq 3$.

