# OpenReview forum: "Does Adversarial Transferability Indicate Knowledge Transferability?"
_ICLR.cc/2021/Conference — Reject_

### Official Review · AnonReviewer2 · 2020-10-28
**Interesting problem, solid analysis, but lack of interpretations and applications**

**Rating:** 5
**Confidence:** 4

**Review:**

##########################################################################
Summary:

This paper demonstrates the relationship between adversarial transferability and knowledge transferability. The authors first formulate these two phenomena. Then under mild assumptions, they prove adversarial transferability indicates knowledge transferability. The authors further justify their theory with experiments.

##########################################################################
Reasons for score:

Overall, I believe this paper studies an interesting problem. The relationship between adversarial transferability and knowledge transferability is of interest to both transfer learning and adversarial attack community. However, the paper does not provide how to further interpret their findings our apply them in practice.

##########################################################################
Pros

The theoretical analysis seems sound and clearly organized. The authors provide a comprehensive analysis of their hypothesis both theoretically and empirically.

##########################################################################
Cons

1.	Although the problem studied is interesting, the results are not surprising. Adversarial transferability indicates the gradients of the two models are similar, which further indicates similar parameters and predictions. Models with similar parameters and predictions will naturally share knowledge. Even without reading the paper, one can expect the argument above will hold.
2.	The authors did not elaborate on how to use their results in applications. For example, since adversarial transferability and knowledge transferability are related, will adversarial training help transfer learning? On the other hand, will transfer learning improve adversarial robustness?

---

> ### Author Response · Authors · 2020-11-13
> **Response to Reviewer #2**
>
> We are glad that you find this problem interesting and we thank you for your valuable time and comments. Regarding your two concerns:
> 1. Results are not surprising: Though the observations are intuitive, we indeed aim to provide rigorous theoretical justification for them and we believe this should be a strength instead of a weakness of our analysis, which certifies our correctness. Please note that establishing this tie rigorously itself is a nontrivial effort.
> 2. Elaboration on applications: Although we mainly focus on our theoretical analyses, based on the experiments we can foresee a few future use cases, which might not be practical for now but surely worth further explorations. Firstly, in the second experiment, we measure transferability between multiple attribute classifiers and an identity classifier. This suggests that the adversarial examples can be used as a proxy to measure the identity classifier’s bias towards certain attributes, such as skin color or gender. This has great implications for model interpretability and fairness research. Secondly, by evaluating the adversarial transferability, we can potentially identify candidates for transfer learning. When the pool of candidate models is large enough, it would be impractical to fine-tune every single one of them. Last but not the least, we can also use knowledge transferability to ensure the adversarial diversity within an ensemble, such that it would be robust and not susceptible to a single kind of attack as an ensemble. Along this line, there are a lot of exciting ideas left to be explored. We hope this will adjust your concerns and start a discussion on potential applications. We will add related discussion in our revision as well.

---

### Official Review · AnonReviewer4 · 2020-10-29
**Shows adversarial transferability is correlated with knowledge transferability but no clear takeaway**

**Rating:** 5
**Confidence:** 3

**Review:**

This paper studies the relationship between adversarial and knowledge transferability. It develops two metrics to measure adversarial transferability and empirically shows that adversarial transferability correlates with knowledge transferability.

Pros:
The paper studies an interesting problem. Intuitively, both adversarial and knowledge transferability show how much the decision boundaries of the source and target classifiers are aligned. This paper is interesting in that it tries to quantify their correlation.

Cons:
The paper provides an interesting observation but fails to further investigate how this observation can be used to gain better understanding of either fields of adversarial examples or knowledge transfer or how the insights can lead to better techniques. In the following, I provide a few comments and questions.

On metrics tau1 and tau2:
-	Why do we need \tau1 and \tau2 as proxies for adversarial transferability? It seems from experimental results that the simple metric of adversarial loss can indeed be a simpler and more robust indicator of the knowledge transferability.
-	\tau1 measures the correlation of the perturbations for the two models. The attack method used finds the perturbation that causes the largest change in output, which is the formulation for the misclassification attack. Different models may classify the perturbed example into different classes and so the output changes would be different. The paper says \tau1 is not enough and we need to look into the changes made to the output as well and whether an affine layer can map one into another. I feel if paper considered targeted attack, the attack could generate highly correlated output changes and so \tau2 wouldn’t be necessary. I wondered if authors tried targeted attack formulation and how the new tau1 would correlate with adversarial transferability.
-	I’m a bit confused about Theorem 1 and how it relates to definition of \tau2. As far as I understand, \tau2 captures how much outputs of two models can be aligned using an affine transformation. Theorem 1 then says large \tau2 means an affine function g can be found that aligns the two models. Isn’t it just the way that \tau2 is defined?

Analysis and results:
-	Some parts of the paper seem a bit disconnected or not elaborated well enough. For example, what do theorems 2 and 3 tell us? Similarly for proposition 2. What are the takeaways?
-	Do experiments confirm the theoretical bounds and derivations? It might be good to define a simple synthetic task (maybe on a logistic regression or 1-layer neural net) to evaluate the claims and show the relationship between adversarial and knowledge transferability.
-	The results need more investigation. For example, in figure 2 (right), \tau1 values are almost identical, which as paper mentions, shows \tau1 is not sufficient for determining transferability. However, although \tau2 values are correlated with transferability, they are almost zero. What does this imply? Do these results statistically support the claims? Similarly, in table 1, the relationship between \tau1 and \tau2 and adversarial-knowledge transfer does not seem to be statistically significant.
-	Some statements in observations seem obvious. For example, section 5.1 reads “we show that the closer the source data distribution is to the target data distribution, the more adversarially transferable the source model to the reference model, thus we observe that the source model is more knowledge transferable to the target dataset.” This is a known concept that has been used in both designing better attacks and also better transfer learning methods.


Paper edit suggestions:
- Writing and paper structure can be polished. I suggest moving background info and definitions into a section background and then laying out the structure for the rest of the paper.
- The discussion of related work and the cited papers need a revision. Some papers are repeatedly cited while others seem to be discussed out of place. For example, page 8 refers to (Zamir et al., 2018) a total of nine times.

---

> ### Author Response · Authors · 2020-11-13
> **Response to Reviewer #4**
>
> Thank you for the valuable comments and your time! Our responses to each of the comments are shown below.
>
> On metrics $\tau_1$ and $\tau_2$:
> * Why do we need $\tau_1$ and $\tau_2$ as proxies for adversarial transferability?: Yes, one can use the adversarial loss as a proxy for adversarial transferability in practice, as we show in the experiments. However, to theoretically analyze the correlation in a clean and precise way, we find $\tau_1$ and $\tau_2$ are surprisingly handy tools to use, and we believe that the two metrics capture the nature of the adversarial transferability. For example, it shows in an elegant form in Theorem 1, where we can derive the equality relationship using $\tau_1$ and $\tau_2$.
> * "I wondered if authors tried targeted attack formulation...":  Thank you for the interesting idea. Intuitively, $\tau_1$ measures the alignment angle between the two perturbations, while $\tau_2$ measures the projected deviation (norm) of the adversarial perturbation in the steepest direction. So both are very important in characterizing the adversarial transferability regardless of targeted or untargeted attack. Nonetheless, you brought up a very interesting idea of using a targeted attack to characterize the transferability. We think it could potentially lead to tighter bounds.
> * Confusion about $\tau_2$: Sorry for the confusion. $\tau_2$ captures how much the deviation of the two models given adversarial attacks can be aligned. That is, $\tau_2$ only uses the information from adversarial attacks. On the other hand, our theorems derive the knowledge transferability on the (non-adversarial) target data distribution.
>
> Analysis and results:
> * "What are the takeaways?": Thanks for pointing this out. Theorem 2 tells us that larger $\tau_1$ and $\tau_2$ indicate potentially small differences of gradients between the target model and the transferred model. With the assumption of smoothness, the small difference of gradients translate to knowledge transferability, which is shown by theorem 3. As for proposition 2, it shows the positive correlation between $\tau_1$ and the adversarial loss. We will discuss more in the revision.
> * "experiments confirm the theoretical bounds": Thanks for the comments, and we do perform experiments to verify our theoretical results in three different transfer learning settings with complex datasets. To make our claim even stronger, we are working on a synthetic experiment and will reply with the results soon.
> * "The results need more investigation": Thanks for the suggestion. As we discuss in the paragraph just before definition 4, adversarial attacks (approximately) only reveal information about the direction with the largest singular value of the Jacobian. That’s being said, the higher the dimension, the less the adversarial attacks can predict the knowledge transferability. As a result, we actually expect to see $\tau_1$ and $\tau_2$ would have worse predicting results for the high-dimensional dataset or high-dimensional models, e.g., BERT on NLP datasets (figure 2 right), and ResNet-18 on CelebA (tabel 1). Thus, it is interesting to see the trend of correlation is still observed. We are currently working on small synthetic experiments to further investigate our theory. We will discuss more in the revision.
> * "Some statements in observations seem obvious": Thanks for the comments, and our goal is to provide theoretical justification for the common observation here which we believe is important for the community. In addition, the example mentioned here is one of the three distinct transfer learning settings. Though it seems obvious, It would be incomplete without it.
>
> Paper edit suggestions: Thanks for the very helpful suggestions, we will make improvements accordingly.

---

### Official Review · AnonReviewer1 · 2020-10-29
**Some concerns need to be solved**

**Rating:** 5
**Confidence:** 4

**Review:**

Summary and contributions
This paper studies the relationship between adversarial transferability and knowledge transferability. By defining two quantities to measure the adversarial transferability, it shows that adversarial transferability measured in this way indicates knowledge transferability both theoretically and empirically.

Strengths
This paper is the first work to theoretically focus on the correlation of two prevalent phenomena of DNN--adversarial transferability and knowledge transferability. Its backgrounds and theoretical results and proofs are clearly presented. Moreover, the experiments are complete: they include experiments for three types of knowledge transferability whose datasets, transfering methods and results are very clear.

Weaknesses
First, this paper does not provide the intuition to use squared cosine value rather than the cosine value in the defination of $\tau_1$(although Proposition 2 theoretically shows the relationship between $\tau_1$ and Cross Adversarial Loss). Fig.1 in the paper also regards $\tau_1$ as the cosine value. For an example of the ill-defined parts of $\tau_1$, given $\bm \delta_{f_1}=\bm \delta_{f_2}=-\bm \delta_{f_3}$, one would claim that the adversarial transferabilities of $f_1$ to $f_2$ and $f_1$ to $f_3$ are same considering their same squared cosine values, which is counterintuitive to make $\tau_1$ an appropriate quantity to measure the similarity between two attacks.

Second, in the definition of $\tau_2$, the authors provide the linear map $A$ and $\tau_2$ without literally stating their physical meanings. Besides, the demonstration of $\tau_2$ in Fig.1 is misleading since it seems that $\tau_2$ is a vector in Fig.1 which is not true. The authors also do not clarify the reasonability of comparing vectors of different dimensions which is necessary for most settings of this paper.

Third, this paper does not provide an indicator with both $\tau_1$ and $\tau_2$ for high adversarial transferability. Futhermore, it lacks the theoretical relation between $\tau_2$ and Cross Adversarial Loss to demonstrate its reasonability.

If these concerns could be solved appropriately, I would consider to raise the score.

Correctness
I have carefully checked the proofs of all theorems in this paper, and ensure that they are correct.

Reproducibility
Yes. The setting of their experiments is clear and complete.

Clarity
The idea, structure and expression of this paper are easy to understand and follow. However, the explanation of Definition 3 is hard to understand.

Relation to the prior work
Yes. Authors show they have a good understanding of prior work's contributions, especially the three types of knowledge transferability.

Addition
Pro. Authors use PGD-attack adversarial transferring in Section 5.1 but virtual adversarial transferring in Section 5.2. Is there any difference between them, and if so, what's the difference?
Pro. Authors claim that $g$ is a trainable function in Section 3 while use a specified $g$ instead in Theorem 3. Does a better $g$, such as $g = \arg \min \left\|g \circ f_{S}-y\right\|_{\mathcal{D}}^{2}$, make the bound in Theorem 3 tighter?

---

> ### Author Response · Authors · 2020-11-13
> **Response to Reviewer #1**
>
> Thank you for your valuable time and constructive comments. Regarding your concerns and questions, our responses follow.
>
> Intuition to use squared cosine value in $\tau_1$: Thanks for pointing it out. This is actually one of the implications we are giving: cosine value being either $-1$ or $1$ has the same indication on high knowledge transferability. This is because fine-tuning the last layer can rectify such difference by changing the sign of the last linear layer. We will discuss more on this in the revision.
>
> Physical meaning of $\mathbf{A}$ in $\tau_2$: $\mathbf{A}$ is the best linear map trying to align the two deviations ($\Delta_{f_1, \delta_{f_1}}$ and $\Delta_{f_2, \delta_{f_1}}$). That’s being said, it characterizes how the deviations of the $f_1$ and $f_2$, given  adversarial attacks, are transferable. Furthermore, $\mathbf{A}$ serves as a guess on the best linear map to align $f_1$ and $f_2$, using only the information from adversarial attacks. In addition, we refer to our response to a similar question about $\tau_2$ raised by reviewer #4 for more discussion. We will make the discussion clearer in the revision.
>
> Indicator with both $\tau_1$ and $\tau_2$ for high adversarial transferability: $\tau_1$ and $\tau_2$ encode and only use the information from adversarial attacks. Thus, they could serve as a definition of adversarial transferability.
>
> Relationship between $\tau_2$ and the adversarial loss: It is interesting to see how $\tau_2$ is related to adversarial loss, and we believe our analysis here can further help the understanding about adversarial transferability for the community.
>
> Difference between PGD-attack and virtual adversarial attack: Sorry for the confusion, they are actually referring to the same thing. The attack method is a projected gradient based attack that uses the network output as ground truth, which is also called virtual adversarial attack.
>
> Does a better $g$ makes bound tighter: Yes, a better $g$ would make the bound tighter. Our bound works as a worst case scenario.

---

### Official Review · AnonReviewer3 · 2020-11-03
**The discovery of this paper may not inspire the community enough as claimed.**

**Rating:** 5
**Confidence:** 3

**Review:**

##########################################################################

Summary:

This paper study the fundamental relationship between adversarial transferability and knowledge transferability. Theoretical analysis is conducted, revealing that adversarial transferability can indicate knowledge transferability. In this procedure, two quantities are formally defined to measure adversarial transferability from different aspects. Furthermore, empirical evaluation in three different transfer learning scenarios on diverse datasets are carried out, showing a strong positive correlation between the adversarial transferability and knowledge transferability.


##########################################################################

Reasons for score:

Strengths:

- This work study the relationship between adversarial transferability and knowledge transferability, which is still under explored.

- Both adversarial transferability and knowledge transferability are defined quantitatively, which enables in-depth understanding of them.

Weaknesses:

- The novelty and contribution of this work are marginal. Although the problem is fundamental, this paper does not seem to be an adequate exploration of the relationship between adversarial transferability and knowledge transferability. The proof of a conclusion, which seems intuitive, may only bring us limited inspiration.

- The three experiments repeatedly prove the positive correlation between the adversarial transferability and knowledge transferability in three knowledge transfer scenarios, which seems to be repetitive. More ingenious experiments can be designed and conducted.

- The organization of this paper is somewhat confusing，e.g. Section 4.

##########################################################################

---

> ### Author Response · Authors · 2020-11-13
> **Response to Reviewer #3**
>
> Thank you for your valuable time and constructive comments. Regarding your three concerns:
>
> The discovery of this paper: We would like to emphasize that the contributions of this work are not only on showing the relationship between the two transferabilities, but also the theoretical framework (e.g., the $\tau_1$ and $\tau_2$) that we build to justify the correlation between adversarial and knowledge transferability. $\tau_1$ and $\tau_2$ are computable, within the range of $[0, 1]$, and encode sufficient information from adversarial attacks to infer transferability (note that in Theorem 1 the result shows with equality). With this framework, we believe that one could explore more about the relationship with less effort. As for inspiration, besides the theoretical foundation for further analysis, we also provide three distinct use cases in the form of experiments. Though they might not be practical yet, we believe they all have tremendous implications, as we will explain in the answer to the following question.
>
> Experiments: Although the three experiments prove the same meta-conclusion, they are in fact quite different in terms of practical settings. In the first experiment, we transfer models between different data distributions, where the input data is drawn from different sources such as sports news and movie reviews. The adversarial transferability here can become a good indicator of the distribution shift in online text processing. In the second experiment, the data distribution is the same, but the label distributions are different for each pre-trained model. Leveraging the adversarial transferability, we can measure common knowledge between the facial recognition model and the attribute classifier. This can be a good indicator of whether facial recognition is biased toward certain attributes, such as skin color or gender, which is a great tool to improve interpretability and verify the fairness of the model. In the third experiment, we explore a harder transfer learning setting, in which even the tasks are different. With adversarial transferability, we plot similar hierarchical clusters between different tasks, which might have great implications for multi-task learnings. We on purpose designed the three experiments to verify the correlation between adversarial and knowledge transferability from different perspectives.
>
> Organization of the paper: Thanks for pointing it out. We will reorganize section 4 to make it more clear for each case in our revision.

---

### Author Response · Authors · 2020-11-25
**Summary of Revisions**

We thank the reviewers for the constructive comments. We have revised the manuscript to incorporate the reviewers’ valuable feedback, and added a set of synthetic experiments to further verify our theoretical findings. Please find the summary of our revision below, and major revisions are highlighted in blue in the updated pdf submission.

1. **On Section 4**: We split the old section 4 and merge part of it into theory section and the rest into experiment section.

2. **On repetitive citations**: We get rid of the repetitive citations on page 8.

3. **On motivations and applications**: We add discussion on potential applications at the beginning of section 4.

4. **Intuitions and takeaways for the theorems**: We added intuitions and provided key takeaways for $\tau_1$, $\tau_2$, as well as Theorem 2, Theorem 3 and Propositions 2.

5. **Synthetic experiments**: We add a synthetic experiment on radial basis regression using one-hidden-layer neural networks.The results exhibit not only the correlations between adversarial transferability and knowledge transferability, but also verify our theoretical results in Theorem 2, Theorem 3 and Proposition 2.

---

### Decision · Program_Chairs · 2021-01-07
**Final Decision**

**Decision:**

Reject

**Comment:**

This paper studies the relationship between adversarial transferability and knowledge transferability. It develops two metrics to measure adversarial transferability and a theoretical framework to justify the positive correlation between adversarial transferability and knowledge transferability. Synthetic experiments show that adversarial transferability measured by the proposed metrics indicates knowledge transferability.

While the paper studies an interesting and fundamental problem, with a sound theoretical analysis and a clear presentation, reviewers still have several reservations to directly accept it.
- Lack of interpretation. How this observation can be used to gain better understanding of either fields of adversarial examples or knowledge transfer?
- Lack of inspiration. How the insights can lead to better transfer techniques, apply to practical applications, and foster future research?
- Lack of justification. Why such definitions of metrics are the intrinsic ways of measuring adversarial transferability? How well do they correlate with the practical experience with advanced attack, defense, and transfer methods?

AC believes the endeavor made by this paper towards a fundamental problem is highly necessary to our field. But given the above reservations, AC would encourage the authors to further strengthen their work to make it more inspiring and useful.